



# Depolymerization and mineralization – investigating N availability by a novel [15]N tracing model

Louise C. Andresen[1], Anna-Karin Björsne[1], Samuel Bodé[2], Leif Klemedtsson[1], Pascal Boeckx[2] and Tobias Rütting[1]

[1]Department of Earth Sciences, University of Gothenburg, Gothenburg, 405 30, Sweden

[2]Isotope Bioscience Laboratory, ISOFYS, Ghent University, Ghent, 9000, Belgium

*Correspondence to*: Louise C. Andresen (louise.andresen@gu.se)

**Abstract.** Depolymerization of soil organic matter such as proteins and peptides into monomers (e.g. amino acids) is currently thought to be the rate limiting step for N availability in terrestrial N cycles. The mineralization of free amino acids

(FAA), liberated by depolymerization of peptides, is an important fraction of the total N mineralization. Accurate assessment of peptide depolymerization and FAA mineralization rates is important in order to gain a better understanding of the N cycle dynamics. Due to the short time span, soil disturbance and unnatural high FAA content during the first few hours after the labelling with the traditional [15]N pool dilution experiments, analytical models might overestimate peptide depolymerization rate. In this paper, we present an extended numerical [15]N tracing model Ntrace which incorporates the FAA pool and related

N processes in order to 1) provide a more robust and coherent estimation of production and mineralization rates of FAAs; 2) and 2) suggest an amino acid N use efficiency ($NUE_{FAA}$) for soil microbes, which is a more realistic estimation of soil microbial NUE compared to the NUE estimated by analytical methods. We compare analytical and numerical approaches for two forest soils; suggest improvements of the experimental work for future studies; and conclude that: i) FAA mineralization might be as equally an important rate limiting step for gross N mineralization as peptide depolymerization rate is, because

about half of all depolymerized peptide N is consecutively being mineralized; and that ii) FAA mineralization and FAA immobilization rates should be used for assessing $NUE_{FAA}$.

**Keywords**: Free amino acids, nitrogen, [15]N, numerical model, microbial nutrient use efficiency, amino acid mineralization, depolymerization rate






## 1. Introduction

Soil organic nitrogen (SON) mineralization is essentially a sequence of depolymerization of polymeric organic compounds followed by mineralization of the liberated monomers (Schimel and Bennett, 2004). Inorganic nitrogen (IN), such as nitrate ($NO_3^-$) and ammonium ($NH_4^+$), as well as free amino acids (FAAs) are known to be the main plant N sources (Schimel and

Chapin, 1996; Bardgett et al., 2003). Therefore, it is essential to know more about the production of mineral and amino acid N, and the balance between mineralization and immobilization of N, in order to have a better understanding of N availability. Gross N mineralization includes mineralization of FAAs, mineralization of other organic monomers and potentially also includes a share of $NH_4^+$ released from mineral complexes (Houlton and Morford 2015). It has been estimated across grassland, cropland and heathland ecosystems that FAA mineralization can be a substantial fraction of the N mineralization

(34 to 88 %; reviewed in Andresen et al., 2015). Hence, amino acid mineralization is important for IN availability.

The microbial N use efficiency (NUE) representing the balance between immobilization and mineralization, is regulated by the soil organic matter (SOM) quality, e.g. soil C to N ratio (Mooshammer et al. 2014). A soil carbon (C) to N (C/N) ratio of 20 is suggested as a breakpoint where NUE reach a maximum (Mooshammer et al. 2014), as a result of microbial retention of N due to N limitation (at high NUE). Contrastingly, high N mineralization leading to low NUE, results

from C limitation (Mooshammer et al., 2014). Carbon or N limitation of a soil is hence thought to govern the direction of the N flow. However, soil C/N ratio is a rather blunt measure for the C/N ratio of substrates used by microbes (potentially camouflaging inert mineral components, or recalcitrant soil organic matter). Schimel and Bennett (2004) suggested to address depolymerization rates, driven by extracellular enzymatic activity, as the rate limiting step for the terrestrial N cycle, hereby determining availability of the amino acid substrate within the soil and assessing the N release from the biologically

available fraction in the soil. Following this, Wanek et al. (2010) provided methodological development of $^{15}N$ pool dilution essays to determine gross peptide depolymerization rates, and by combining this with $^{15}N$ tracing, quantification of gross FAA mineralization can in addition be achieved (Andresen et al., 2015). These approaches apply analytical calculations (Kirkham and Bartholomew, 1954; Watkins and Barraclough, 1996) handling one flux at the time, which has some obvious limitations (Rütting et al. 2011). To advance our understanding of the organic N dynamics and mineralization, we deem it

timely to present a novel numerical $^{15}N$ tracing model.

In this paper we combine, for the first time, two parallel $^{15}N$ tracing experiments, in which soil is separately amended with $^{15}N$ labelled ammonium or an amino acid mixture. For data analysis, we further developed the numerical $^{15}N$ tracing model *Ntrace* (Müller et al. 2007) to explicitly account for FAA turnover, in order to simultaneously quantify gross peptide depolymerization, gross FAA mineralization and total gross N mineralization in forest soils. With this approach a

more robust and coherent rate assessment and a more accurate calculation of microbial amino acid nutrient use efficiency ($NUE_{FAA}$) is achieved. Furthermore, we discuss the importance of FAA mineralization for gross N mineralization and



present the peptide depolymerization and FAA mineralization rates as two important steps co-limiting for N availability in forest soils.

## 2. Methods

### 2.1 Field site

Soil was sampled from two forests at the Skogaryd Research Catchment part of SITES (Swedish Infrastructure for Ecosystem Studies, www.fieldsites.se), situated in southwest Sweden (58° 23'N, 12° 09'E; 60 m above sea level). Mean annual temperature is 6.4 °C and the mean annual precipitation is 709 mm (Ernfors et al. 2011). The soil of the first forest was an Umbrisol with sandy loam texture and was planted with Norway spruce (*Picea abies*) in the 1950s. The vegetation was classified as a spruce forest of low herb type based on the classification system by Påhlsson, 1998, with sparse ground

vegetation dominated by bryophytes (*Mnium hornum*, *Polytricum formosum* and *Pleurozium schreberi*). The second forest was on a Podzol soil, where the vegetation was classified as a spruce forest of bilberry type (Påhlsson, 1998). The tree stand (Norway spruce) was 55-130 years old and of 23-30 m height. The ground vegetation was dominated by *Vaccinium myrtillus* and mosses.

### 2.2 Soil sampling

Soil was sampled with an auger on 14[th] April 2014 (Umbrisol) and 12[th] May 2014 (Podzol), each with five field replicates. The air temperature at both sampling times was 11° C. For the Umbrisol, the thin litter layer and vegetation was pushed aside and the soil was sampled until 10 cm depth. For the Podzol, the soil was sampled below the O-horizon and 10 cm down. These depths were selected to get matching low SOM contents. The soil was immediately transported to the lab, where roots and stones were manually removed. Wet soil (40 g) was placed in 250 mL glass bottles with a lid with a small hole, and pre-

incubated for a week at constant temperature (20 °C) prior to labelling with $^{15}$N.

### 2.3 $^{15}$N labelling incubations

The pre-incubated soil was labelled with $^{15}$N in two different treatments, either receiving ($^{15}$NH$_4$)$_2$SO$_4$ (99 % $^{15}$N) or $^{15}$N-amino acid mixture ('Cell Free' amino acid mix, 20 AA, U-$^{15}$N 96-98 %, chemical purity >98 %, Cambridge Isotope laboratories, USA). The total N addition with NH$_4^+$ was 0.6 µg N g$^{-1}$ dry soil. The total added amino acids (AA) was 9.32 µg

N g$^{-1}$ dry soil (Umbrisol) or 7.72 µg N g$^{-1}$ dry soil (Podzol). The label solution was added into the pre-incubated soil using a pipette (4 ml per bottle) and quickly stirred with a clean spatula.

  Soils from NH$_4^+$ labelling was extracted using a 1 to 2 soil to liquid ratio, with 1 M KCl, by shaking for one hour at 120 rpm, then the samples were filtered (Whatman qualitative filter papers, No 1) and kept frozen (-18° C) until further



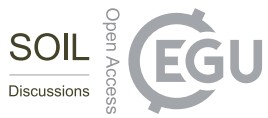

processing. Soil extractions were done on one set of samples directly after labelling (13 min), the rest of the bottles were incubated in a dark room at constant temperature (20º C) until extraction after 24, 48, 96, 168 and 240 hours (h).

Soils from AA labelling were divided in two parts immediately after label addition, prior to incubation in the dark room. Simultaneously one half was extracted with 1 M KCl as described above, and the other half was extracted with 10 mM

$CaSO_4$ containing 3.4% formaldehyde, in 1:2 soil to liquid ratio, by shaking for 1 h at 120 rpm, then the samples were filtered (Whatman qualitative filter papers, No. 1) and kept frozen until further processing. Extractions were done 13 min and 0.5, 1, 2 and 6 h after labelling.

## 2.4 Analysis of $^{15}$N

All KCl extracts were prepared for analysis of $^{15}$N contents of $NH_4^+$ by chemical conversion to $N_2O$ and analyzed by IRMS

(ANCA-TGII interfaced with a 20-20 IRMS, SerCon, UK) as described by Stevens and Laughlin (1994). $NH_4^+$ concentration was measured using a flow injection analyzer (Auto Analyzer 3, Bran+Luebbe, Norderstedt, Germany).

The $CaSO_4$ extracts for $^{15}$N–AA analysis were purified using cation-exchange cartridges (OnGuard II H, 1 cc, Dionex), conditioned with ultrapure water (>18.2 MΩ), 3 M $NH_3$ and 1 M HCl. After loading the extract on the cation-exchange resin, the cartridge was washed with 10 mL ultrapure water and AAs were eluted with 30 mL 3 M $NH_3$. The

purified sample was dried under reduced pressure at 35 °C, and finally derivatized using ethanol-pyridine and ethylchloroformate (Husek et al., 2001). Finally the individual FAAs were measured by gas chromatography – mass spectrometry (GC – MS, Trace GC – DSQ, Thermo Fisher). Separation was done on a OV1701 column (30 m × 0.25 mm ID × 0.25 μm film; Sigma-Aldrich, Diegem, Belgium) for the 16 amino acids: Alanine and Glycine (Ala and Gly; m/z: 116/117), Valine (Val; m/z: 144/145), Leucine, Serine, Isoleucine and Threonine (Leu, Ser, Ile and Thr; m/z: 158/159),

Proline and Aspartic acid (Pro and Asp; m/z: 142/143), Asparagine (Asn, m/z: 188/189), Methionine (Met, m/z: 249/250), Glutamic acid (Glu; m/z: 202/203), Phenylalanine (Phe; m/z: 192/193), Lysine (Lys; m/z:156/157), Tyrosine (Tyr; m/z: 280/281) and Tryptophan (Trp; m/z: 130/131). The following amino acids were not possible to measure: Arginine, Glutamine, Histidine and Cysteine.

## 2.5 Soil properties

Soil water content was determined gravimetrically (GWC) by oven drying of c. 10 g soil samples to constant weight at 75 °C. Soil pH was measured in 1 M KCl extracts. Soil organic matter was determined on 2 g of dried soil samples by loss of ignition (8 h at 500 °C). Total soil N and C content was determined on ground soil with an elemental analyzer (ANCA SerCon, Crew, UK).





## 2.6 Calculations

### 2.6.1 Analytical equations

In order to quantify the gross N mineralization ($M$), $NH_4^+$ consumption ($C_{NH4}$), FAA consumption ($C_{FAA}$) and peptide depolymerization rates ($D_{SON}$; Fig. 1) the analytical equations developed for isotope pool dilution experiments were used
(Kirkham and Bartholomew, 1954; Andresen et al., 2015):

For $p > c$:

$$p = \frac{M_t - M_0}{t} \times \frac{\ln(H_0 M_t / H_t M_0)}{\ln(M_t/M_0)} \tag{1}$$

$$c = \frac{M_t - M_0}{t} \times \frac{\ln(H_0/H_t)}{\ln(M_t/M_0)} \tag{2}$$

For $p = c$

$$c = p = \frac{M_{av}}{t} \times \ln\left(\frac{H_0}{H_t}\right) \tag{3}$$

Where   $p$ = production rate (i.e. $M$ or $D_{SON}$; respectively), $c$ = consumption rate (i.e. $C_{NH4}$ or $C_{FAA}$, respectively), $M_i$ = total content of labelled pool, $H_i = {}^{15}N$ content of labelled pool. Indices $i$ indicate: initial ($0$), final ($t$) and average ($av$) content.

In order to quantify the FAA mineralization rate ($M_{FAA}$; Fig. 1), the following equations were used (Watkins and
Barraclough, 1996; Andresen et al., 2015):

$$M_{FAA} = \quad p * \frac{a'_1 *(M_t/M_0)^{\frac{p}{\theta}} - a'_0}{a'_{aa} *(M_t/M_0)^{\frac{p}{\theta}} - a'_{aa}} \tag{4}$$

With $p$ being gross mineralization rate obtained from Eq. (1) or Eq. (3) from the ${}^{15}N\text{-}NH_4^+$ labelling experiment, extrapolated to 0-6 h by a logarithmic function; $\theta$ is $(M_t-M_0)/t$ with $M = NH_4^+$ content from the ${}^{15}N\text{-}AA$ labelling; $a'_{aa}$ is the excess ${}^{15}N$ fraction of the total FAAs pool averaged for the two time steps; $a'$ is the excess ${}^{15}N$ fractions of the $NH_4^+$ pool from the ${}^{15}N\text{-}$
AA labelling.

### 2.6.2 Iterative numerical model *Ntrace*

Numerical ${}^{15}N$ tracing models have been used to investigate soil inorganic N dynamics (Myrold and Tiedje, 1986; Rütting et al., 2011). Among the main advantages of a numerical approach is that process specific gross N transformation rates are quantified simultaneously rather than sequentially (Rütting and Müller, 2007). Therefore, interactions between N
transformations are accounted for. Here we further developed the ${}^{15}N$ tracing model *Ntrace* (Müller et al., 2007) to explicitly include FAA dynamics (Fig. 1). The mineralization of complex soil organic matter is represented as a two-step process: 1)



peptide depolymerization releasing free AAs (FAA) (depolymerization rate $D_{SON}$), and 2) mineralization of FAA to $NH_4^+$ (amino acid mineralization rate $M_{FAA}$). In addition, mineralization of other (non-peptide OR non-AA-polymers) SON ($M_{SON}$) to $NH_4^+$ was included, which accounts for depolymerization followed by mineralization of other N compounds (e.g. chitin; Bai *et al.*, 2013). Gross N mineralization is hence the sum of $M_{FAA}$ and $M_{SON}$. Immobilization of FAAs ($I_{FAA}$) and ammonium

($I_{NH4}$) is also included in the model. In the current study, $^{15}NO_3^-$ could not be measured even after addition of $^{15}NO_3^-$, due to too low $NO_3^-$ content of the soil. Therefore, oxidation and immobilization of $NH_4^+$ could, not be separated and the quantified gross $NH_4^+$ immobilization ($I_{NH4}$) is the sum of these two processes. The N transformations were either implemented as zero-order kinetics for infinite substrate pools ($D_{SON}$ and $M_{SON}$) or first-order kinetics for finite pools ($M_{FAA}$, $I_{FAA}$ and $I_{NH4}$).

A Markov chain Monte Carlo sampling was used for parameter estimation by fitting the model to measured contents

and $^{15}N$ enrichments of the studied pools (Müller et al., 2007). The outcome is a probability density function for each model parameter, from which parameter averages and standard deviations can be calculated (Rütting and Müller, 2007). For $D_{SON}$ in the Podzol, the probability density function was truncated at zero. Therefore, average and standard deviation for that parameter were calculated using functions for truncated normal distributions (Cohen and Woodward, 1953; Cicchinelli, 1965). For N transformations described by first-order kinetics, average gross rates were calculated by integrating the gross

rates over the experimental period. A good fit of the model to the experimental data was achieved (Figs. 2 and 3).

A mix of 20 different amino acids was added to the soil. However, four of the added AAs (Arginine, Cysteine, Glutamine and Histidine) could not be measured with the current methodology. The N of these four AAs accounted for 22 % of the added $^{15}N$ in the experiment. As those AAs also contribute to the mineralization ($^{15}N-NH_4^+$ production), these were considered in the tracing model as follows: we assume that the soil pool of non-measured AAs has the same average $^{15}N$

enrichment as the pool of the measurable 16 AAs. The pool of non-measured AAs was then included, having the same depolymerization, mineralization and immobilization rates as the measured AAs. In order to evaluate the potential effect of the assumption of the same $^{15}N$ enrichment, an uncertainty data analysis with altered $^{15}N$ enrichment for the missing AAs was conducted, which indicated that altered $^{15}N$ enrichment had only marginal effects on the estimated gross rates (see Supplement Table S1). We argue that the most realistic gross rates are quantified when including the non-measured AAs.

However, in order to compare the results from the *Ntrace* with the analytical rates, $M_{FAA}$ and $I_{FAA}$ were additionally calculated for the measured AAs only, either for the entire incubation period ('240 h') or for the first 6 h only (same time-frame as for analytical calculations). The gross rates including all AAs will be higher in proportion to the amount of AA-N (i.e. 22, when compared to rates for measurable AAs only).

### 2.6.3 Nitrogen use efficiency

Microbial N use efficiency of free amino acids (***NUE***; Fig. 1) is the fraction of consumed FAAs that is not released as ammonium but incorporated into the microbial biomass (Mooshammer et al., 2014). We calculated ***NUE_FAA*** based on the *Ntrace* results as:



$$NUE_{FAA} = I_{FAA} / (I_{FAA} + M_{FAA}) \tag{5}$$

For results from analytical solution, Mooshammer et al. (2014) calculated **NUE** as:

$$NUE = (C_{FAA} - M) / C_{FAA} \tag{6}$$

Equation 6 implies that gross N mineralization derived from the analytical calculations is solely derived from FAA mineralization.

## 3. Results and discussion

### 3.1 Soil properties

Both investigated soils were acidic with pH of 3.7, but differed in other properties (Table 1). The Umbrisol had higher SOM and total C and N content, but lower C/N ratio. Nitrate concentration was below detection limit for both soils. The Umbrisol had prior to the [15]N addition a six-times higher FAA content compared to the Podzol and the relative abundance of individual FAAs differed as well between the two soils (Fig. 4). The FAA composition in the soil was initially dominated by acidic or non-aromatic compounds; possibly other FAAs might have been removed from the soil solution through plant root or microbe uptake (Andresen et al., 2011, Chen et al., 2015).

### 3.2 Analytical versus numerical approaches for quantification of gross N rates

To our knowledge, quantification of total gross FAA mineralization and peptide depolymerization rates had not been done by means of numerical tracing models until now. However, these models represent robust methods to asses gross transformation rates, as all data points from the two isotope label experiments and all observed time steps are included. A particular weakness of analytical approaches is that substrate addition stimulates consumption processes (Schimel, 1996; Di et al., 2000), which is also true for FAAs. This problem can be minimized by using numerical tracing models as the stimulation will be greatest immediately after [15]N labelling, but numerical models allow integration of transformation rates over a much longer period (Rütting et al., 2011). We indeed found that the numerical derived gross rates for $M_{FAA}$ and $I_{FAA}$ when integrated over 6 h, were several fold higher than rates integrated over the entire experimental duration (240 hours, Table 2; Fig. 5). Differences between gross rates derived from analytical and numerical models were greatest for FAA consumption, while smaller differences were found for $D_{SON}$ and total mineralization (Table 2; Fig. 5). This points to an over stimulation of the processes by addition of FAAs, and demonstrate the advantage of longer incubation time with numerical data analysis to achieve more realistic gross rates.

From the Podzol we observed both FAA mineralization ($M_{FAA}$) as well as mineralization of other SON ($M_{SON}$). The total gross N mineralization rate ($M_{FAA} + M_{SON}$) derived from *Ntrace* integrated over 240 h was lower, but comparable to the analytically determined gross mineralization (*M*) rate (Table 2; Fig. 5). In this soil, FAAs mineralization contributed by only





12 to 15% to total gross N mineralization. For the Umbrisol, gross mineralization from *Ntrace* was only half the gross rate estimated by the analytical model and entirely assigned to $M_{FAA}$. The analytical model does not separate between mineralization from FAA or other N forms ($M_{FAA}$ or $M_{SON}$), but provides one total rate ($M$). Quantification of FAA mineralization is possible using the analytical Eq. (4), which though requires a $^{15}$N tracing approach and two $^{15}$N labellings

(FAA and $NH_4^+$). The analytical derived $M_{FAA}$ (data not presented, Eq. 4) was in both soils higher than $M$ (Eq.1 or 3), which is irrational. This might have been caused by the different time frames or the stimulation of $M_{FAA}$ but not of $M$ by the AA addition. In any case, numerical $^{15}$N tracing models overcome such inconsistencies, as all gross rates are quantified simultaneously.

Depolymerization rates ($D_{SON}$) quantified by *Ntrace* were smaller compared to the analytical results for the Podzol,

but these were similar for the Umbrisol (Table 2; Fig. 5). Gross depolymerization quantified by the analytical approach only had a minor decrease with increasing incubation time (Table 2; Fig. 5 c, d), suggesting no or only little re-mobilization of $^{15}$N (Bjarnason, 1988). The similarity of $D_{SON}$ rates quantified with *Ntrace* and by analytical approach confirms the validity of the numerical tracing model. The main difference between the two approaches is that the numerical approach estimates the rate for the entire 240 h of incubation, while the analytical approach considers a limited time span of max. 6 h.

**3.3 Gross N dynamics in two contrasting forest soils**

As the numerical *Ntrace* model is less prone to disturbance by $^{15}$N label addition and as interactions between different N transformations are taken into account, we suggest that this approach provides more realistic gross N transformation rates. The ratio of total gross N mineralization ($M$) to peptide depolymerization ($D_{SON}$) rate ranges from 5 to 25 % in organic soils, based on analytical calculations (Wanek et al., 2010; Wild et al., 2015). We found in both soils much higher ratio, being 76

% for Umbrisol and 170% for Podzol using the analytical approach, while *Ntrace* resulted in $M$ to $D_{SON}$ ratios of 46 % and 400 %, respectively. Thus, gross N mineralization was highly important for the N cycle and for making N available in the soil. Moreover, the $M_{FAA}$ amounted to 46% (Umbrisol) and 65% (Podzol) of $D_{SON}$ (Table 2; *Ntrace* Fig. 5). The finding of c. 50 % of depolymerized peptide N being further mineralized to $NH_4^+$ as well as the higher total mineralization than peptide depolymerization in the Podzol, suggest that peptide depolymerization is not the single major rate limiting step for the soil N

cycle (Schimel and Bennett, 2004). Rather the results suggest that amino acid mineralization rate can be a co-limiting step for plant N availability in terrestrial ecosystems.

The Podzol was characterized by a lower peptide depolymerization rate compared to previously studied sub- and top-soils from forests and grasslands (Wild et al., 2015). The Umbrisol soil, being more N, SOM and FAA rich (Table 1), showed consistently higher gross N transformation rates (Table 2; Fig. 5). This agrees with the finding of a correlation of

high N status and faster N cycling (organic and inorganic) across cold-temperate forests (Finzi and Berthrong, 2005). One pronounced difference between the two soils was the mineralization dynamics: for the Umbrisol the gross N mineralization was estimated as entirely derived from the FAA pool (100 % $M_{FAA}$), while in the Podzol $M_{FAA}$ contributed only by 15 % to




the total gross N mineralization (Fig. 5). Consequently, the Umbrisol strongly depended on FAAs as source for IN, while in the Podzol the mineralization of other organic N forms ($M_{SON}$) dominated the IN production. Notably, $M_{FAA}$ was about ten-fold lower in Podzol compared to Umbrisol. This can be explained by the much smaller $D_{SON}$ in the Podzol (Fig. 5), limiting the substrate for $M_{FAA}$ in this soil, which is also reflected in the six-fold smaller FAA content (Table 1). Variation in the

contribution of $M_{FAA}$ to $M$ has been reviewed previously, ranging from 35 % to 100 % across agricultural and natural soils, from results obtained using analytical calculations (Andresen et al., 2015).

The observed differences in gross N transformation rates are connected to differences in soil organic matter quality and properties of the microbial biomass (Farrell et al., 2014). The C/N ratios for the two investigated soils were near the breakpoint (C/N ratio of 20) suggested by Mooshammer et al. (2014), at which a change from C limitation to N limitation of

the microbial community occur (Fig. 6). By using the gross rates from *Ntrace*, the **$NUE_{FAA}$**s were 0.57 for Umbrisol and 0.60 for Podzol, which is smaller than expected from the relationship presented by Mooshammer et al. (2014) (Fig. 6). However, the *Ntrace* derived **$NUE_{FAA}$s** agree with the results from the analytical approach obtained from the longest time step (30 mins to 360 mins), but not for the shorter time steps (Table 2; Fig. 6). For Umbrisol the **$NUE_{FAA}$** from the analytical approach (Eq. 6) at the shorter time steps (30 min to 60 min and to 120 min) were higher and fell within the confidence interval from

Mooshammer et al. (2014; Fig. 6). We account this to the fact that Eq. (6) uses gross FAA consumption rates quantified by the analytical approach. As it is well understood, this approach provides an overestimation of consumption rates ($C_{FAA}$), due to substrate addition (Schimel, 1996; Di et al., 2000), hereby, the **$NUE$** (Eq. 6) will be biased towards high values. The Podzol showed significant input to gross mineralization from other organic N than FAAs therefore, the **$NUE$** of Podzol derived from the analytical equation (Eq. 6) (time step 30 min to 120 min) was low. Consequently, **$NUE_{FAA}$** is ideally

assessed by considering FAA mineralization explicitly (Eq. 5). If the true **$NUE_{FAA}$** is lower as we suggest from the *Ntrace* approach, it is likely that a larger portion of FAAs taken up by microbes is subsequently mineralized, than would be suggested from the line in Fig. 6. This challenges the understanding of the shift of soil C limitation to N limitation, however the two investigated soils can neither be termed as N or C limited.

### 3.4 Suggested improvements of the laboratory method

Following the recommendation by Wanek et al. (2010), the FAA label addition was 10 to 20 times larger than the initial FAA content in the original substrate (litter or soil). This requires pre-knowledge of the FAA content in the soils. The addition of FAAs might cause an unintended 'hot-spot' effect (Kuzyakov and Blagodatskaya 2015) which stimulates depolymerization, by priming, and this is difficult to avoid in such an experimental approach (Schimel, 1996; Di et al., 2000). Addition of a too small amount of FAAs would, however, potentially give enrichments of the individual FAAs at or

below the detection limit and should be avoided. Therefore, future studies could apply lower amounts of FAA, thereby further avoiding an unwanted stimulation of gross N rates.





Adsorption (physical-chemical) of the added label to SOM cannot be evaluated with our methodology, even when comparing: initial FAA, added FAAs and the FAA amount after 10 min time step (data not shown), because significant microbial N-transformations cannot be excluded (Jones et al., 2013). However, during the first time steps (30 min, 60 min and 120 min) only little change in $^{15}$N % fraction was observed (Fig. 2 and 3), suggesting quite small depolymerization. This

point to the fact that there can be a limit to how small a peptide depolymerization rate we can measure with the current methodology. The individual FAAs were consumed equally through the time series as suggested by decreases in content (data not shown), and at the final time step, the individual FAA contents were back at the background level, hence, our procedure encompass a life-cycle for the added FAA quantity.

We did not assess peptide depolymerization rates for individual FAAs, because  transformations between FAAs

(Knowles et al., 2010) can neither be ruled out nor tested with our experimental set up (e.g. potential aspartic acid formation from asparagine or break down of larger FAAs such as lysine to smaller size such as serine). We aimed at quantifying gross rates relevant for organic N transformations in soils, using incubations with either $^{15}$N-NH$_4^+$ or $^{15}$N-AA mix label. Only the first sampling time point was synchronized for the two incubation types, we suggest that at least one more synchronized sampling (e.g. after 6 h) should be done in future experiments. Furthermore, after addition of $^{15}$N labelled FAA, we observed

a $^{15}$N enrichment of NH$_4^+$ even at the last extraction (Fig. 2, 3). Therefore, future studies should include later extractions (e.g. at 48 h) to follow the fate of the added $^{15}$N-AA. We expect that these suggestions would further improve the model estimation.

Finally, a further improved understanding of the FAA dynamics can be achieved by improving the analytical capacity for measuring all 20 proteinogenic FAAs. In this paper we could not measure 4 of the added amino acids, but they

were considered in *Ntrace* for realistic quantification of especially $M_{FAA}$. Assumptions had to be made for the $^{15}$N enrichment of the non-measured FAAs and that the non-measured FAA concentration decreased like the measured FAAs. The quantified gross rates were not very sensitive to alteration of the $^{15}$N enrichment of the non-measured FAAs (see Suppl. Table). We assumed similar behavior as all 16 measured FAAs showed similar time courses in soil content after labelling (data not shown).

An even further development of the numerical model would include the NO$_3^-$ dynamics in a soil with that property. Another outlook is that depolymerization rates of polymers other than amino acids (such as chitin) are potentially an important part of the total depolymerization. Hence, further research is needed to uncover the importance of other limiting steps in the N-cycle.

Conclusively, we suggest that i) numerical modelling in conjunction with $^{15}$N tracing should be used when

determining FAA mineralization, peptide depolymerization and gross N mineralization rates as a preferred alternative to the analytical equations; ii) FAA mineralization and FAA immobilization rates are used to determine microbial $NUE_{FAA}$ as this gives a better estimation of NUE than if NUE is based on gross N mineralization (M) and C$_{FAA}$; iii) FAA mineralization



might be as equally an important rate limiting step for gross N mineralization as peptide depolymerization rate is, because about half of all depolymerized peptide N is consecutively being mineralized and iv) depolymerization of other components in the soil is an additional potentially rate limiting step for the N cycle, which needs further investigation.

**Author contribution**

TR, LK, PB and LCA planned the experiments; AKB and LCA conducted the field work and lab-incubations, supervised by TR. SB conducted the stable isotope analysis. TR developed the *Ntrace* model. All authors discussed the conceptual model and contributed to data interpretation and the writing of the paper.

**Acknowledgements**

Stijn Vandevoorde and Katja van Nieuland from ISOFYS are thanked for analysis of amino acids, inorganic nitrogen and
soils. The project was financial supported by the Swedish Research Council Formas and the Swedish strategic research area "Biodiversity and Ecosystem services in a Changing Climate – BECC" (www.becc.lu.se/).

**Supplement Table S1.**

Sensitivity analysis of different initial $^{15}$N values for the 'missing' amino acid pool.

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



**Tables**

Table 1. Soil properties for the two soil types: Podzol and Umbrisol from Skogaryd. Averages with standard error. pH (in 1 M
5  KCl) Gravimetric soil water content (GWC), Soil organic matter (SOM), dry soil carbon (C) and nitrogen (N), C to N ratio of dry
soil, total free amino acid content (FAA).

|  | Podzol | Umbrisol |
|---|---|---|
| pH | 3.7 ± 0.0 | 3.7 ± 0.0 |
| GWC (%) | 34.2 ± 2.1 | 52.9 ± 3.8 |
| SOM (%) | 6.9 ± 0.4 | 9.3 ± 0.8 |
| Dry soil C (%) | 3.4 ± 0.2 | 4.7 ± 0.5 |
| Dry soil N (%) | 0.15 ±0.01 | 0.24 ± 0.03 |
| C to N ratio | 22.7 ± 0.4 | 19.4 ± 0.3 |
| FAA ($\mu$g g$^{-1}$DW) | 1.3 ± 0.6 | 7.7 ± 3.1 |





**Table 2. N dynamics rates from analytical equations (Eq. 1, 2, 3, 4 and 6) and *Ntrace* numerical model, average in [nmol g$^{-1}$ h$^{-1}$] and standard deviation. Peptide depolymerization rate ($D_{SON}$); FAA immobilization rate ($I_{FAA}$); FAA mineralization rate ($M_{FAA}$); amino acid consumption ($C_{FAA}$); mineralization rate of organic N ($M_{SON}$); gross N mineralization ($M$); ammonium consumption ($C_{NH4}$); immobilization rate of NH$_4^+$ ($I_{NH4}$) and microbial amino acid nutrient use efficiency ($NUE_{FAA}$) (Eq. 5 and 6). For $D_{SON}$,**
5 **$C_{FAA}$, and $NUE_{FAA}$ the time step 30 to 360 min is presented; $M_{FAA}$ and $I_{FAA}$ for all 20 AAs over 240 h. $C_{FAA}$ and $M$ from *Ntrace* are calculated sums ($M = M_{SON} + M_{FAA}$ and $C_{FAA} = I_{FAA} + M_{FAA}$).**

|  | Podzol | | Umbrisol | |
| --- | --- | --- | --- | --- |
|  | **Analytical** | ***Ntrace*** | **Analytical** | ***Ntrace*** |
| $D_{SON}$ | 4.2 (3.8) | 1.3 (0.9) | 22.6 (10.8) | 20.6 (2.9) |
| $I_{FAA}$ | - | 1.2 (0.1) | - | 12.3 (1.3) |
| $M_{FAA}$ | - | 0.8 (0.1) | - | 9.4 (0.9) |
| $C_{FAA}$ | 22.4 (2.9) | 2.0 (0.2) | 60.8 (13.7) | 21.7 (2.2) |
| $M_{SON}$ | - | 4.4 (0.4) | - | 0.0 |
| $M$ | 7.2 (2.5) | 5.2 (0.5) | 17.1 (9.7) | 9.4 (0.9) |
| $C_{NH4}$ ($I_{NH4}$) | 3.6 (0.8) | 4.1 (0.4) | 11.9 (14.2) | 9.2 (1.4) |
| $NUE_{FAA}$ | 0.62 (0.39) | 0.60 (0.12) | 0.61 (0.86) | 0.57 (0.12) |



**Figures**

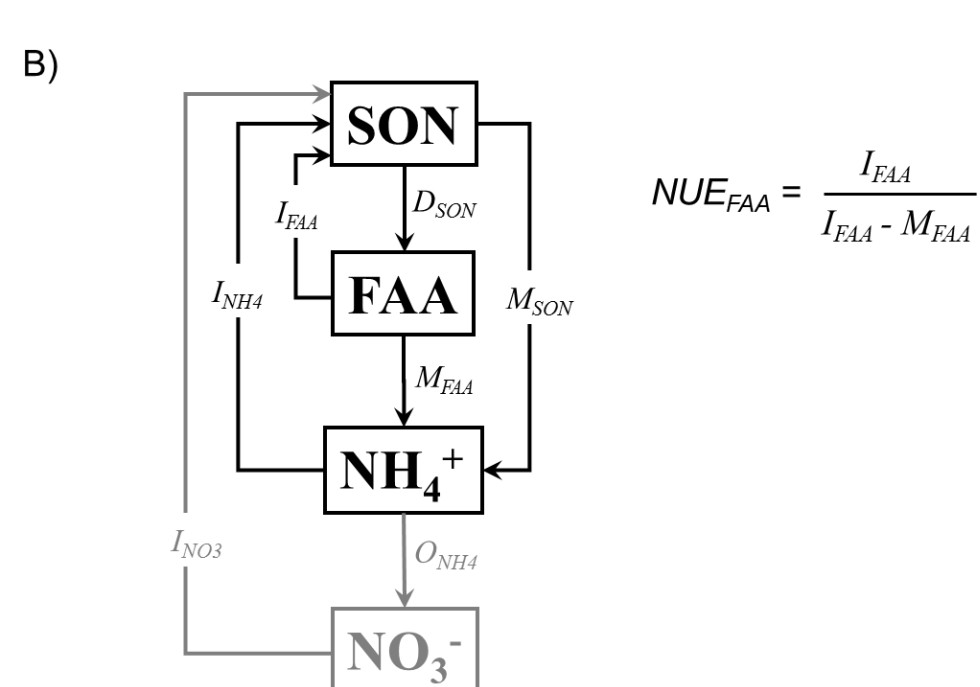

5   **Figure 1. A) Schematic analytical model; Gross N mineralization ($M$); ammonium consumption ($C_{NH4}$); amino acid consumption ($C_{FAA}$) and peptide depolymerization rate ($D_{SON}$); B) The conceptual model *Ntrace* considers pools for: soil organic nitrogen (SON), free amino acid (FAA), ammonium ($NH_4^+$) and nitrate ($NO_3^-$), and fluxes of peptide depolymerization rate ($D_{SON}$), FAA mineralization rate ($M_{FAA}$), FAA immobilization rate ($I_{FAA}$), mineralization rate of organic N ($M_{SON}$), immobilization rate of $NH_4^+$ ($I_{NH4}$), $NH_4^+$ oxidation rate ($O_{NH4}$) and $NO_3^-$ immobilization rate ($I_{NO3}$). Grey pools and fluxes could not be investigated in the**
10  **current study due to too low nitrate content.**




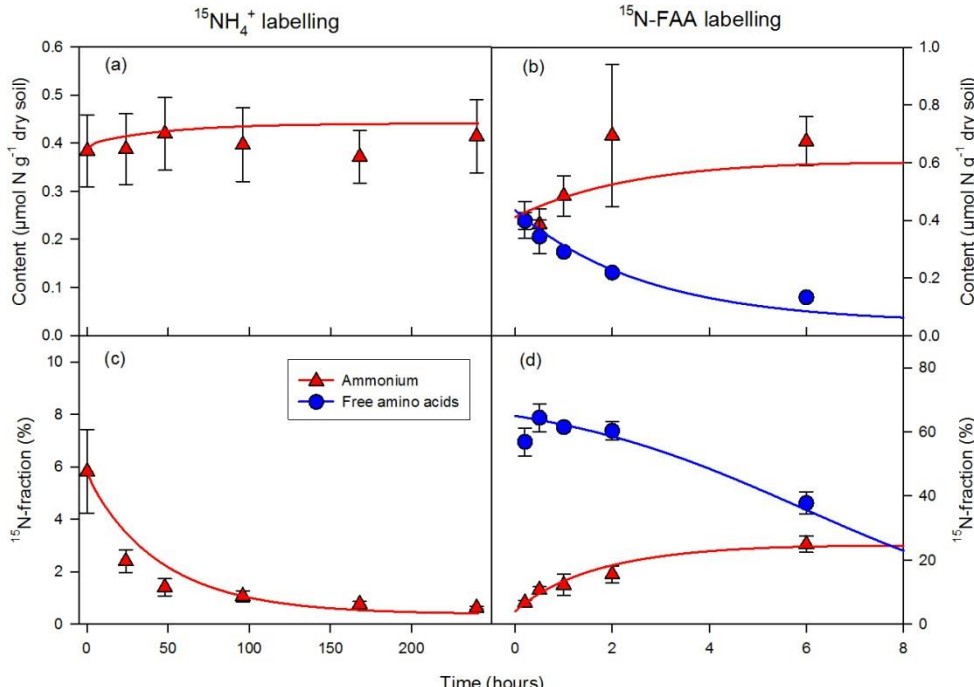

**Figure 2. Umbrisol, time flow of the two labelling experiments: $^{15}$N-NH$_4^+$ labelling and $^{15}$N-FAA labelling; symbols indicate data observation with standard error, and lines indicate the two AA-pool model, where triangles and red is ammonium and circle and blue is FAAs. (a) and (b) N content [µg N g$^{-1}$ DW soil] and (c) and (d) $^{15}$N fraction (%).**



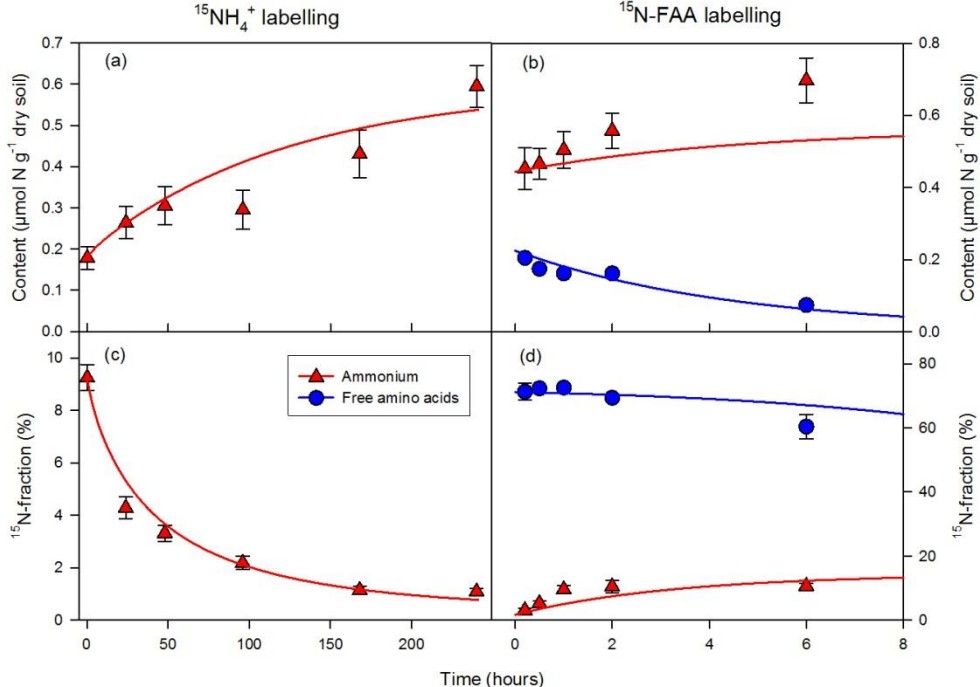

**Figure 3. Podzol, time flow of the two labelling experiments: $^{15}$N-NH$_4^+$ labelling and $^{15}$N-FAA labelling; symbols indicate data observation with standard error, and lines indicate the two AA-pool model, where triangles and red is ammonium and circle and blue is FAAs. (a) and (b) N content [μg N g$^{-1}$ DW soil] and (c) and (d) $^{15}$N fraction (%).**



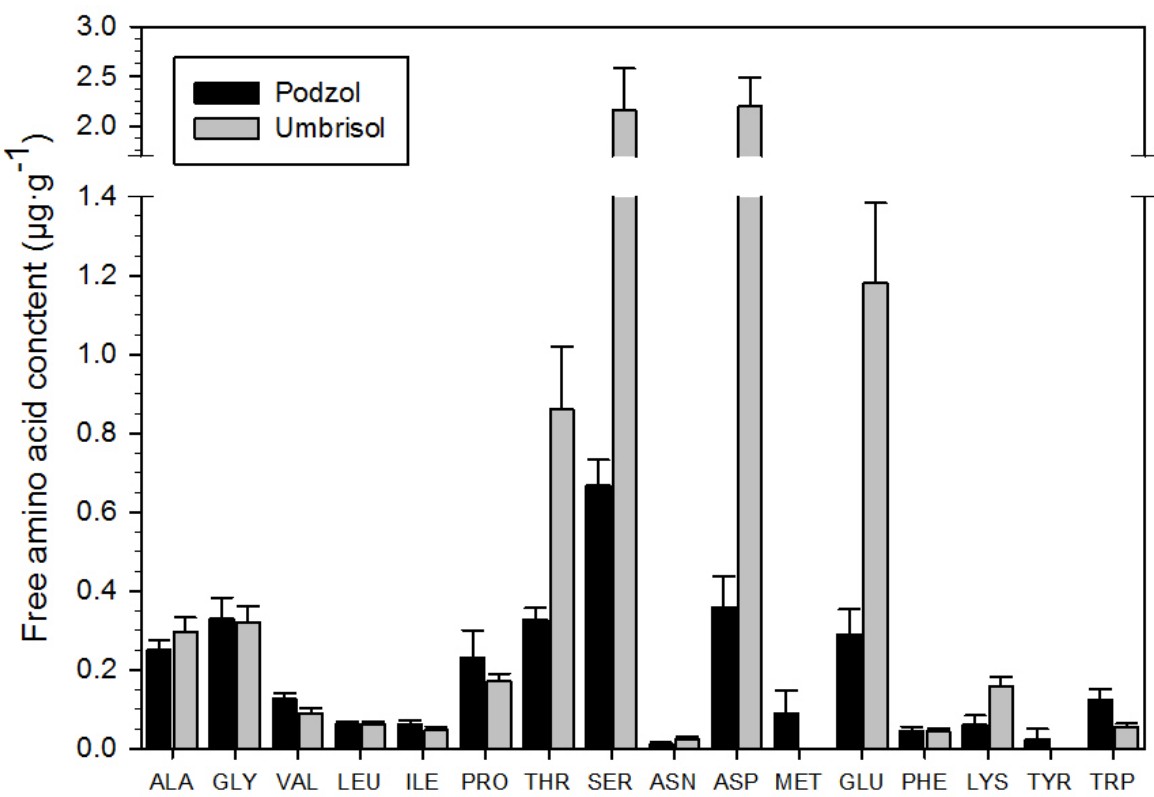

**Figure 4. Initial soil content of individual amino acids (μg FAA g$^{-1}$ DW soil).**



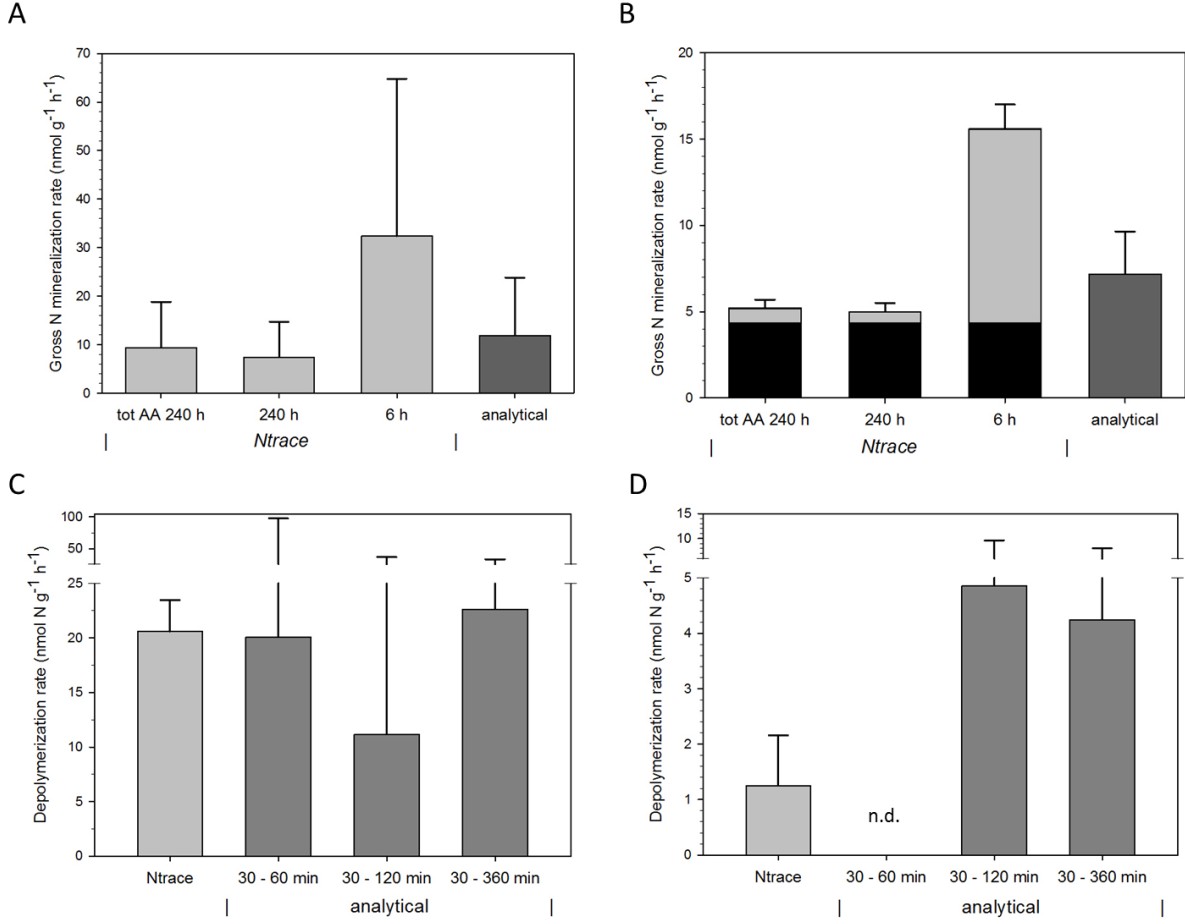

**Figure 5. N transformation rates obtained by numerical modelling (*Ntrace*) and analytical equations. Gross N mineralization rates [nmol N g$^{-1}$ h$^{-1}$] for Umbrisol (A) and Podzol (B); from the *Ntrace* model as sum of MSON (dark; note in Umbrisol $M_{SON}$ is zero) and $M_{FAA}$ (light grey): 'tot AA 240 h' is calculated for all 20 AAs over 240 h; '240 h' is calculated for the 16 measurable AAs over 240 h and '6 h' is calculated for the 16 measurable AAs for the initial 6 h; 'analytical': (dark grey) from the analytical equation for the time step 0 to 24 h. Depolymerization rate (total) in [nmol N g$^{-1}$ h$^{-1}$] as average with standard deviation, for Umbrisol (C) and Podzol (D); from *Ntrace* model; or at the time steps: '30 - 60 min' (for Podzol this is not determined (n.d.) due to unchanged $^{15}$N in all replicates), '30 - 120 min' and '30 - 360 min' from the analytical.**




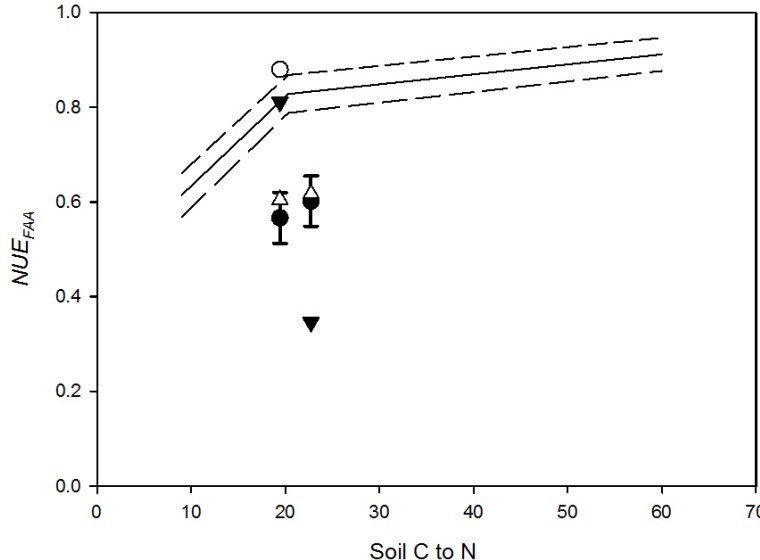

**Figure 6.** $NUE_{FAA}$ **(microbial amino acid nutrient use efficiency) seen in relation to the soil C to N ratio (19.4 for Umbrisol and 22.7 for Podzol);** $NUE_{FAA}$ **calculated by Eq. (5) upon numerical model calculation is filled circles; open circle is NUE from analytical calculated rates of AA consumption at time steps 30 to 60 min and gross N mineralization extrapolated to 6 h, using Eq. (6), for Podzol this was not determined due to unchanged** [15]**N in all replicates; filled triangles at time steps 30 to 120 min; and open triangle at time steps 30 to 360 min. The two-pieced line of NUE vs. soil C to N ratio from Mooshammer** *et al.* **2014 is regression line with standard error, from organic- and mineral soils and plant litter calculated from Eq. (6).**