# Peer review of "Simultaneous quantification of depolymerization and mineralization rates by a novel 15N tracing model"

_SOIL, 2016_

## Referee Comment (RC1) · Anonymous Referee #1 · 25 Mar 2016

In the manuscript "Deploymerization and mineralization – investigating N availability by a novel 15N tracing model" by Louise C. Andresen et al. the authors compare an analytical approach and a numerical model based on experimental 15N tracing to estimate soil N dynamics. The main N processes discussed are gross rates of protein depolymerization and NH4 mineralization. The authors also calculate microbial nitrogen uses efficiency from the results of the analytical and the numerical approach. The presented manuscript is clearly written, structured well and the results are presented in a meaningful manner. The approach to compare an analytical method and a numerical tracing model is also of great interest. However there are two major concerns at this point: 1. In contrast to the repeatedly mentioned studies by Schimel and Bennett and

Mooshammer et al., the authors of this study do not include microbial biomass as an explicit pool in their model. While neither Schimel and Bennett nor Mooshammer et al. use a numerical model, the concept of a microbial pool can change the interpretation of the results. Without knowing the details of the presented model in this study, I assume that the results might look very different when a microbial pool with a specific turnover time and dynamics is included. This might specifically question the importance and the interpretation of the differences between the two investigated soils regarding the Mson flux in the Ntrace model (see detailed comments). At the very least the authors should discuss how an explicit microbial pool might change their model. 2. The much more severe problem with the experimental setup is the enormous input of amino acids. Although the authors argue that Ntrace is better suited to deal with this surplus of free amino acids, by integrating over a longer time than the analytical methods, the exact opposite could be argued. By introducing this flush of amino acids, peptidases could be inhibited by their potential product and peptidase expression could be down-regulated due to the surplus of amino acids, which would result in lower depolymerization rates later in the incubation. This could be an explanation for the low depolimerization rates derived from Ntrace compared to the analytical approach. It is also not clear to me why the authors chose these high amino acid amendments. The method described by Wanek et al. was developed for leaf litter, which can be expected to have much higher FAA concentrations. Also Wanek et al. mention twice in their paper that FAA concentrations should be determined beforehand and only 25% of the amino acid pool should be amended to avoid the effects the authors of this study discuss. The drastic change in the amino acid pool by these high amendments might thus bias both the analytical and the numerical approach. Neither of the approaches might thus represent realistic N dynamics in the investigated soils. I am afraid the only way to overcome this problem and to sustain the current line of argumentation is to repeat the experiment with lower amendments of amino acids.

Specific comments: Abstract: Page 1, lines 16-17: while stated here and repeatedly throughout the manuscript, that the numerical approach is superior to the analytical

method, later in the manuscript (page 8, lines 12-13) it is argued that the numerical model is valid because it produces results for Dson that are similar to the analytical approach. This is contradictory. Introduction: The Introduction should be concluded with concrete, testable hypotheses, similar to those presented in the abstract. These hypotheses should be revisited in the discussion. Page 2, lines 23-24: Please state some of these obvious limitations. Page 2 Lines 26-27: This sentence should be in past tense.

Methods: Page 7, lines 4-5: The underlying concept of a microbial N pool in Mooshammer et al. however also allows for the interpretation that any changes in these dynamics might be caused by changes in the microbial N pool. This interpretation is not possible with the Ntrace model.

Results and discussion: Page 7, lines 16-17: As mentioned above, the integration over a longer time does however not consider any physiological adaptations of the microbial community to the amino acid flush. Page 7, lines 17-19 and Page 9, lines 25-31: This problem has been addressed by Wanek et al., who suggested to determine the FAA pool and only amend 25 % of that pool. Page 7, line 24: Since both of the presented approaches have their limitations and are biased by the large amount of amended amino acids, I think it is not possible to tell which method is more realistic. It would be interesting to compare the numeric model with the analytical approach and lower amino acid amendments. This might also help to evaluate if it is necessary to include an explicit microbial N pool in models for soil N dynamics. Page 8, lines 5-6: This might again be caused by the large amounts of amended amino acids. Page 8, Lines 21-26: When microbes are included in the interpretation of these results, it could also mean, that the addition of amino acids led to an increased uptake of amino acids but also an increased release of excess N as NH4 from the microbial biomass. Together with a potential down-regulation of peptidase activity this could be the reason for the observed results. Page9,lines 1-6: In this model the Mson pathway is relevant, when changes in the NH4 pool cannot fully be described by the changes in the FAA pool.

[Figure]

If a microbial pool was included in the model, changes in this pool, which should be situated between FAA and NH4 could be responsible for the observed dynamics. Page 10, lines 12-14: Especially for the amino acid pool dilution method a longer incubation time of 6h might result in problems with recycled labelled N. Figures: Please stat for all figures that include error bars what these are and what the sample size was. Figure 1: The second formula for NUE should have IFAA+MFAA in the denominator.

---

## Referee Comment (RC2) · Anonymous Referee #2 · 30 Mar 2016

There has been increased interest in the cycling of organic N in soils. This manuscript provides a quantitative assessment of amino acid production and consumption in soils using a 15N tracing model approach that builds upon prior work of senior author.

General comments: 1. Table 1. It is peculiar that soil ammonium concentration is not provided. If available (and it should be), it should be added. It is important for the reader to know how the amount of 15N-label added compares to the natural background. 2. Units. Although maybe not the best, most 15N tracer studies provide concentrations and rates in terms of mass of N (e.g., mg, ug) rather than as moles N. I would suggest tables and figures be converted to mass of N to make the data easily comparable to previous studies. Also, rates are most often "per day" rather than "per hour". 3.

[Figure]

Model. It strikes me as odd that SON was not separated into microbial biomass and non-biomass pools. Any N "immobilized" into the non-biomass pool will be misrepresented as N assimilated by the microbial biomass and thus misrepresent NUE as the term is commonly understood. Along with this, measuring the 15N incorporated into the microbial biomass (e.g., using chloroform fumigation) would have been a helpful addition. 4. Model comparisons. I don't know that there is a "right" way, but comparing rates from a zero-order analytical model with a mixed kinetic numerical model seems fraught. The attempt at determining an integrated rate for a given time period seems reasonable, but exactly how this was done is not described in much detail. In reality the rates given by the two model types agree quite well (Table 2) except for Cfaa, which leads me to question the validity of Eqn. 4. I was too lazy to go back to check Barraclough's derivation, but it makes me wonder if there is a flaw in Eqn. 4. Could the averaging to get the a' values be a factor?

Specific corrections: p. 1, l. 8—Use commas to set off the "such as" phrase. I also wonder if "monomers" is too limiting? It is was the authors measure in this research; however, others (Farrell et al. I believe) have shown that small oligopeptides are preferred over amino acids. p. 1, l. 15—delete extra "2)" p. 2, l. 15—a good "classic" reference that would fit here is: "Jansson, S.L., and J. Persson. 1982. Mineralization and immobilization of soil nitrogen. p. 229–252. In F.J. Stevenson (ed.) Nitrogen in Agricultural Soils. ASA, Madison, WI." p. 2, l. 17—delete "to address" p. 2, l. 19—"thereby" rather than "hereby" p. 2, l. 31—define "NUEfaa" p. 3, l. 9—parentheses around the year p. 3, l. 15—the "th" by dates can be deleted p. 3, l. 24—capitalize Laboratories p. 4, l. 4—Provide the rationale for the calcium sulfate/formaldehyde extraction. It is not a standard method that readers will know. p. 4, l. 18-24—amino acids don't need to be capitalized p. 4, l. 25—it is more typical to oven dry mineral soils at 105C; 75C is more normal for plant tissue (or organic soil horizons) p. 5, l. 5—The Andresen ref is inappropriate here as the equations come directly from the original Kirkham and Bartholomew paper. p. 5, l. 18—Are you sure it is logarithmic, or was it exponential? p. 6, l. 6—delete comma after "could" p. 6, l.

8—I don't think the pool is "infinite"; it is large and changes imperceptibly during the short incubation period. p. 6, l. 28—add "%" after "22" p. 7, l. 15—This seems like a "throw-away" sentence as it doesn't really lead to greater understanding of the results. p. 7, l. 18—The problem of additions stimulating processes is as true regardless of what approach one uses to analyze the data. Now, it is true that if the model uses first-order kinetics, then this "mass dependency" is accounted for to some degree, but one can incorporate these kinetics in the analytical model (see "Case 2" in the rarely quoted 1955 paper by Kirkham and Bartholomew). p. 7, l. 16—correct to "assess" p. 8, l. 6—Is "irrational" the best word? The result is illogical, but that raises a question as to whether there is a flaw in the logic behind equation 4? p. 10, l. 5—change to "points" p. 10, l. 26—Chitin is not an amino acid, it is an amino sugar polymer. References—Capitalization of titles is inconsistent (particularly older references)

---

## Referee Comment (RC3) · Anonymous Referee #3 · 6 Apr 2016

In the present manuscript "Depolymerization and mineralization – investigating N availability by a novel 15N tracing model", Andresen et al. compared an analytical and a numerical approach for estimating gross rates of peptide depolymerization, amino acid immobilization, and (amino acid) N mineralization in two forest soils. Current knowledge about soil organic N dynamics is particularly limited, in part due to technical challenges, and thus studies of amino acid turnover are essential to our understanding of N cycling. All methods and models have their limitations, and further improvement of such analytical approaches is paramount for more accurate estimations of gross N dynamics in soils. In the current manuscript, the authors propose that their numerical approach provides more robust and coherent estimations of gross production and con-

sumption rates compared to the "traditional" analytical approach developed by Kirkham and Bartholomew (1954). However, the work in its present form is not convincing, as it has a fundamental experimental flaw and inconclusive reasoning.

My main concern relates to the high amount of 15N-labelled AA to both soils: approx. 600% and 120% of the initial FAA pool was added as 15N-AA to the Podzol and Umbrisol, respectively (based on data given in Table 1). It is known that addition of a given substrate often stimulates its consumption, and thus it is common practice to add only a small fraction of the initial target pool when performing isotope pool dilution assays, in order to minimize such bias on gross rate estimates. Furthermore, the authors erroneously state "Following the recommendation by Wanek et al. (2010), the FAA label addition was 10 to 20 times larger than the initial FAA content in the original substrate (litter or soil)" (Pg 9 L25-26). However, Wanek et al. (2010) clearly state that a maximum of 25% of the initial FAA pool should be added in the form of 15N-tracer, based on a preliminary determination of the size of the FAA pool. The excessive addition of 15N-AA in the present study likely resulted in biased gross rate estimates, in particular CFAA, and thus also affected the calculation of nitrogen use efficiency (NUE). The stimulation of amino acid mineralization rates due to high 15N-AA label addition, are also likely to have resulted in an overestimation of MFAA rates with the numerical approach (and not only rates estimated by the analytical approach), which would explain the high MFAA rates estimated. In Figure 2d and 3d, it becomes also evident that the 15N-AA label was not homogenously distributed, otherwise there would have not been an increase in 15N/14N in the first time interval. Of course, for this time interval, it is not possible to calculate a rate using the analytical approach. The weak dilution of the 15N-AA label during the first time intervals was not due to low depolymerization rates, as suggest by the authors, but rather likely due to high 15N-labelled substrate addition, which resulted in an enrichment of the FAA pool of 60-70 at%. In line with the comment by Referee #1, I thus think that it is necessary to repeat the experiment with appropriate 15N-labelled substrate addition.
This brings me to the question of how to evaluate which approach yields more realistic rate estimates, which certainly cannot be concluded by simply comparing rates estimated by the two different approaches. I believe that both analytical and numerical models have disadvantages and limitations. For instance, I agree with Referee #1 and #2 that the current numerical model is missing a microbial biomass pool with a different turnover time than the SON pool. The authors also claim that their numerical model overcomes the problem of high label addition by integrating gross rates over a longer time. As already pointed out by Referee #1, the enormous 15N-AA input could result in end-product inhibition of peptidases, leading to lower depolymerization rates during longer incubations. Here, I would like to add that longer soil incubations often result in ammonium accumulation over time, due to the absence of plant roots, which would constantly remove a part of the soil nutrients. For example, such an ammonium accumulation was observed for the Podzol, as the ammonium concentration increased 3-fold during the 240h incubation (Figure 2a). Therefore, I suggest that the authors need to provide some experimental evidence that the numerical model actually overcomes problems such as high label addition. Regarding the comparison between the two different approaches, statistical support should be provided for differences in rates estimated by the two models.

Furthermore, the authors claim that using gross rates computed by their numerical model (IFAA and MFAA) yields more accurate estimates of microbial nitrogen use efficiency compared to the model by Mooshammer et al. (2014). The authors raise two points: (1) the analytical approach yields less accurate rate estimates (but see comment above), and (2) Mooshammer et al. (2014) use gross N mineralization (M) instead of amino acid mineralization (MFAA). However, the nitrogen use efficiency model proposed in the present study (NUEFAA) is conceptually different than that by Mooshammer et al. (2014) (NUE). The authors estimate here amino acid-N use efficiency (NUEFAA) based on amino acid immobilization and amino acid mineralization, whereas Mooshammer et al. based their model on gross amino acid consumption rate as proxy for microbial organic N uptake, since proteins are the main N-containing

compounds in soil and plant litter. Therefore, both NUEFAA and NUE models are conceptually justified and there is no evidence that using MFAA instead of M yields better estimates of microbial NUE. Indeed, estimates of microbial NUE could be improved by including also organic N compounds other than amino acids, which, however, remains a great analytical challenge.

I also suggest that the authors should more carefully prepare the manuscript, as there are some inconsistencies and mistakes. Equations of the analytical approach (Eq. 1, 2 and 3) are partially wrong: equation 1 for production rate has a mistake in the numerator of the second term; equation 2 for the consumption rate is actually the production rate; and, equation 3, for the case when there is no change in concentration over time, also seems wrong, at least when compared to their former work (Andresen et al. 2015). Furthermore, the authors used different units: $\mu$g N/g, $\mu$g FAA/g, $\mu$mol N/g, nmol N/g. In some instances, it is not even stated whether it refers to N or FAA (Table 1, Table 2). I suggest using consistent units throughout the manuscript. As I understood it, the initial FAA concentrations presented in Table 1 should correspond to the sum of FAA presented in Figure 4. For Podzol, Table 1 shows 1.3 $\mu$g, whereas in Figure 4 the values roughly sum up to 3 $\mu$g. In turn, for Umbrisol, both Table 1 and Figure 4 seems consistent: 7.7 $\mu$g in Table 1, as well as roughly 7.7 in Figure 4.

Specific comments:

Pg 1 L14: Comma is missing before which

Pg 1 L16: Delete "2)"

Pg 2 L13: correct to "reaches"

Pg 2 L14: From a biological rather than mathematical perspective, I would say that low NUE leads to high N mineralization, and not that high N mineralization leads to low NUE.

Pg 2 L23-24: The authors should state what the obvious limitations are.

Pg 4 L1-2: For N mineralization, 13 min for equilibration between added and native ammonium is quite short.

Pg 4 L3-4: Why were FAAs also extracted with 1 M KCL?

Pg 4: The authors used the protocol developed by Wanek et al. (2010). However, the authors do not even once cite this work in Materials & Methods.

Pg 4 L16: The reference is wrong. It should be Husek, 1991.

Pg 4 L18-23: The authors should explain why some amino acids have the same m/z: Alanine and Glycine (m/z 116/117); Leucine, Serine, Isoleucine and Threonine (m/z 158/159); Proline and Aspartic acid (m/z 142/143);

Pg 5 L5 and L15: The original references (Kirkham and Bartholomew, 1954; Watkins and Barraclough, 1996) are sufficient. Delete reference Andresen et al., 2015.

Pg 5 L13: Specify that it is excess 15N abundance.

Pg 6 L15: I do not always see a good fit of the model in Figure 2 and 3. For example in Figure 3b, the fit of the model for ammonium concentration seems not to fit the experimental data.

Pg 8 L18-19: In Wanek et al. (2010) the samples were plant litter and not organic soil.

Figure 1 B: Equation is wrong.

Table 1: The initial soil ammonium concentrations of both soils should be stated. There is no need to say that the C:N ratio refers to dry soil.

Figure 4: Typo on the y-axis.

References

Andresen LC, Bode S, Tietema A, Boeckx P, Rütting T. (2015). Amino acid and N mineralization dynamics in heathland soil after long-term warming and repetitive drought. SOIL 1: 341-349.

Kirkham DON, Bartholomew WV. (1954). Equations for following nutrient transformations in soil, utilizing tracer data. Soil Sci Soc Am J 18: 33–34.

Mooshammer M, Wanek W, Hämmerle I, Fuchslueger L, Hofhansl F, Knoltsch A et al. (2014). Adjustment of microbial nitrogen use efficiency to carbon:nitrogen imbalances regulates soil nitrogen cycling. Nat Commun 5.

Wanek W, Mooshammer M, Blöchl A, Hanreich A, Richter A. (2010). Determination of gross rates of amino acid production and immobilization in decomposing leaf litter by a novel N-15 isotope pool dilution technique. Soil Biol Biochem 42: 1293–1302.

Watkins N, Barraclough D. (1996). Gross rates of N mineralization associated with the decomposition of plant residues. Soil Biol Biochem 28: 169-175.

---

## Author Comment (AC1) · 27 May 2016

Gothenburg, 27[th] May 2016

Dear Editor and Referees of SOIL Discussions

We thank you for the many constructive comments and corrections to our manuscript. We have used the input to improve the text. The details of our response and revision are given below.

First of all, we have clarified that the main aim of the paper is to present the new version of the $^{15}$N tracing model *Ntrace*. The main advantage of this approach is the simultaneous quantification of rates in a more comprehensive model concept. For this reason we change the title to: 'Simultaneous quantification of depolymerization and mineralization rates by a novel $^{15}$N tracing model', and the abstract is more focused written now:

'**Abstract**. Depolymerization of soil organic matter, such as proteins and (oligo-)peptides into monomers (e.g. amino acids) is currently considered to be the rate-limiting step for nitrogen (N) availability in terrestrial ecosystems. Mineralization of free amino acids (FAA), liberated by depolymerization of peptides, is an important fraction of total mineralization of organic N. Hence, accurate assessment of peptide depolymerization and FAA mineralization rates is important in order to gain a better process-based understanding of the soil N cycle. In this paper, we present an extended numerical $^{15}$N tracing model *Ntrace*, which incorporates the FAA pool and related N processes in order to provide a more robust and simultaneous quantification of depolymerization and gross mineralization rates of FAAs and soil organic N. We discuss analytical and numerical approaches for two forest soils; suggest improvements of the experimental work for future studies; and conclude that: i) FAA mineralization can be an equally important rate limiting step for total gross N mineralization as peptide depolymerization rate, when about half of all depolymerized peptide N is directly mineralized; and that ii) gross FAA mineralization and FAA immobilization rates can be used to develop FAA use efficiency ($NUE_{FAA}$), which can reveal microbial N or C limitation.'

Some of the comments were general across more than one referee (R) and these are treated together (I, II, II and IV below) in a joint **reply to all referees**:

**I. Including microbial biomass**: all three reviewers have an opinion about this matter: **R3**: *'the current numerical model is missing a microbial biomass pool with a different turnover time than the SON pool'*; **R1**: *'the authors of this study do not include microbial biomass as an explicit pool in their model'*; **R1**: *'Methods page 7, lines 4-5: The underlying concept of a microbial N pool in Mooshammer et al. however also allows for the interpretation that any changes in these dynamics might be caused by changes in the microbial N pool. This interpretation is not possible with the Ntrace model'*, **R1**: *'It would be interesting to compare the numeric model with the analytical approach at lower amino acid amendments. This might also help to evaluate if it is necessary to include an explicit microbial N pool in models for soil N dynamics.'*, and **R1**: *'page 8, lines 21-26: When microbes are included in the interpretation of these results, it could also mean, that the addition of amino acids led to an increased uptake of amino acids but also an increased release of excess N as NH4 from the microbial biomass. Together with a potential down-regulation of peptidase activity this could be the reason for the observed results. Page 9, lines 1-6: In this model the MSON pathway is*

*relevant, when changes in the NH4 pool cannot fully be described by the changes in the FAA pool. If a microbial pool was included in the model, changes in this pool, which should be situated between FAA and NH4 could be responsible for the observed dynamics.* ' **R2**: *'It strikes me as odd that SON was not separated into microbial biomass and non-biomass pools. Any N "immobilized" into the non-biomass pool will be misrepresented as N assimilated by the microbial biomass and thus misrepresent NUE as the term is commonly understood. Along with this, measuring the 15N incorporated into the microbial biomass (e.g., using chloroform fumigation) would have been a helpful addition'.* **R3**: *'I agree with Referee #1 and #2 that the current numerical model is missing a microbial biomass pool with a different turnover time than the SON pool.'*

**Our reply**: True, we have not included microbial biomass explicitly in our conceptual model and not in the conducted experimental work. We see some challenges in incorporating a microbial N pool in the model and are not convinced that it will enhance the robustness of gross rate quantifications. The main challenge is that there exists no solid method to quantify active microbial biomass. The problem with the chloroform fumigation method (that R2 suggests) is that an extractability factor must be used in order to come to a value for the microbial biomass. This factor is in reality variable in different soils and soil depths and different extractants (Jörgensen and Müller, 1996). Historically, the factor for nitrogen ($K_{EN}$) is obtained by calibrating against the parallel factor for carbon ($K_{EC}$) (Jörgensen, 1996), which was originally calibrated with an incubation yielding a $CO_2$ measure from (inoculated) soil respiration. We believe, due to the uncertainty of $K_{EN}$ and the chloroform fumigation method, that adding the microbial biomass to the model would complicate the set up unnecessarily and add uncertainty that later would be amplified in the model. Davidson et al. (1991) have stated that: "*As an alternative method, a non-linear equation is given for calculating the gross immobilization rate from the appearance of $^{15}N$ in chloroform-labile microbial biomass; but incomplete extraction of biomass N may result in low estimates*".

Even if we could get a good measure of how many microbes are abundant in the soil, we would not know how many are active in assimilating N. Most of them are probably not active at all (Vandewerf and Verstraete, 1987). Therefore, to just measure the microbial biomass and incorporate it in the *Ntrace* model (like in model 1) would not improve the quantifications but in contrast lead to erroneous results.

In fact, although microbial N was measured in Mooshammer et al. (2014) they did not include microbial N in their model, neither in the calculations of gross rates or NUE.

**In reply to R2:** The immobilization is from our point of view, in our experimental setup, identical to the assimilation in microbial biomass, which due to turnover of microbes then becomes non-living SON (microbial residue N). The only other option in a closed soil incubation without plant roots, is adsorption to soil particles.

**Again, R1:** *'When microbes are included in the interpretation of these results, it could also mean, that the addition of amino acids led to an increased uptake of amino acids but also an increased release of excess N as NH4 from the microbial biomass..... At the very least the authors should discuss how an explicit microbial pool might change their model.'*

**Our reply:** To answer this comment we should look at the concept of nitrogen mineralization (Fig. I). Monomeric organic N (particularly FAA) is taken up by the microorganisms. A part of that N is used in the biosynthesis of microbial biomass, while the N exceeding the demand of biosynthesis is liberated as $NH_4^+$ inside the cell and then exuded to the soil solution. This $NH_4^+$ is though not mixed with the microbial N and consequently will still carry the same $^{15}N$ enrichment as the N taken up.

[Figure]

Figure I. Microbial mineralization of FAA (modified from Reddy & DeLaune, 2008).

Do we need to consider microbial N pool in a $^{15}N$ tracing model to realistically represent this dynamics? For evaluating that, we compare to models (Fig. II) for a situation assuming the following soil N contents (realistic proportions, arbitrary units): $[NH_4^+] = 20$; $[FAA] = 5$ and $[N_{mic}] = 500$. Labelling with $^{15}N$ enriched amino acids leads to a $^{15}N$ enrichment of FAA of 20 %. We assume that1 FAA is taken up, of which half is assimilated (immobilized) and half mineralized. In the following we illustrate with this example the reason we think model 1 is erroneous.

[Figure]

Figure II. Two models for mineralization.

Model 1 – Microbial: the FAA taken up by microbes is 20 % enriched in $^{15}$N. If 1 FAA is taken up, the overall enrichment of the microbial N (prior to mineralization) is 0.04 % [= (1*0.2)/500]. In this model the mineralized N will originate from the microbial N pool and, hence, the $NH_4^+$ released by mineralization will have a $^{15}$N enrichment of 0.04 %. This results in a $^{15}$N enrichment of soil $NH_4^+$ of 0.001 % [= (0.5*0.0004)/20].

Model 2 – Implicit (as *Ntrace*): In that case, the mineralization of FAA directly transfers FAA-N to $NH_4^+$, and the $^{15}$N enrichment of mineralized N will be the same as for the FAA, that is 20 %. This gives a $^{15}$N enrichment of soil $NH_4^+$ of 0.5 % [= (0.5*0.2)/20].

Model 2 considers the fact that the mineralized FAA does not go through the microbial biomass and that the released $NH_4^+$ will still carry the $^{15}$N enrichment of the FAA. This is consistent with the situation that is actually occurring, as illustrated in Fig. II. For this reason, model 2 is a more realistic scenario. As shown, considering a microbial N pool would lead to an erroneous quantification of gross mineralization. Theoretically, explicitly considering and measuring microbial N could improve the quantification of N immobilization, but as sated above this is hindered methodological issues on measuring the microbial N (see also Davidson et al., 1991).

**II. Addition of $^{15}$N-enriched amino acids. R1**: *'The much more severe problem with the experimental setup is the enormous input of amino acids'*, **R1**: *'By introducing this flush of amino acids, peptidases could be inhibited by their potential product and peptidase expression could be down-regulated due to the surplus of amino acids, which would result in lower depolymerization rates later in the incubation.'*, and **R1**: *'It is also not clear to me why the authors chose these high amino acid amendments. The method described by Wanek et al. was developed for leaf litter, which can be expected to have much higher FAA concentrations. Also Wanek et al. mention twice in their paper that FAA concentrations should be determined beforehand and only 25% of the amino acid pool should be amended to avoid the effects the authors of this study discuss.'*, and *'Results and discussion: Page 7 lines 16 to 17: as mentioned above, the integration over a longer time period does however not consider any physiological adaptations of the microbial community to the amino acid flush. Page 7, lines 17-19 and Page 9, lines 25-31: this problem has been addressed by Wanek et al., who suggested to determine the FAA pool and only amend 25% of that pool.'; and 'Page 8, lines 5-6: This might again be caused by the large amounts of amended amino acids'. And **R3**: 'My main concern relates to the high amount of 15N-labelled AA to both soils: approx. 600% and 120% of the initial FAA pool was added as 15N-AA to the Podzol and Umbrisol (based on data given in Table 1). It is known that addition of a given substrate often stimulates its consumption, and thus it is common practice to add only a small fraction of the initial target pool when performing isotope pool dilution assays, in order to minimize such bias on gross rate estimates. Furthermore, the authors erroneously state "Following the recommendation by Wanek et al. (2010), the FAA label addition was 10 to 20 times larger than the initial FAA content in the original substrate (litter or soil)" (Pg 9 L25-26). However, Wanek et al. (2010) clearly state that a maximum of 25% of the initial FAA pool should be added in the form of 15N-tracer, based on a preliminary determination of the size of the FAA pool. ........ As*

*already pointed out by Referee #1, the enormous 15N-AA input could result in end-product inhibition of peptidases, leading to lower depolymerization rates during longer incubations. Here, I would like to add that longer soil incubations often result in ammonium accumulation over time, due to the absence of plant roots, which would constantly remove a part of the soil nutrients. For example, such an ammonium accumulation was observed for the Podzol, as the ammonium concentration increased 3-fold during the 240h incubation (Figure 2a).'.*

**Our reply**: We agree that the high amount of added amino acids was far from ideal, a fact that we already mentioned in the paper (p.9, line 30). We regret that we have wrongly quoted the Wanek et al. 2010 paper. Indeed, the authors recommend a maximum 25% addition of amino acids, related to the background amino acid content. Prior to the experiment we had knowledge on the abundance of amino acids from previous soil samplings in the same site and based on this we chose the amount for amino acid addition. However, it turned out that the soil samples used for the experiment had lower amino acids content than expected, leading to the high additions.

We have re-written the section '3.4 Suggested improvements of the laboratory method' to correctly express the recommendation by Wanek et al. The following sentence is deleted: 'Following the recommendation by Wanek et al. (2010), the FAA label addition was 10 to 20 times larger than the initial FAA content in the original substrate (litter or soil).' And replaced by these sentences: 'The FAA label addition was 10 to 20 times larger than the initial FAA content in the soil. Wanek et al. (2010) recommend adding maximum 25 % of the background amino acid content, but we were not able to reach the recommended level. This specification requires pre-knowledge of the FAA content in the soils.' This is in the section '3.4 Suggested improvements of the laboratory method', and is followed by this text: 'The addition of FAAs might cause an unintended 'hot-spot' effect (Kuzyakov and Blagodatskaya 2015) which stimulates depolymerization by priming (Schimel, 1996; Di et al., 2000). On the other hand, upon addition of high amount of amino acids, peptidases could be repressed (Vranova *et al.* 2013; Glenn *et al.* 1973). Therefore, future studies should apply lower amounts of FAA.

Furthermore, we delete this text: 'The analytical derived *MFAA* (data not presented, Eq. 4) was in both soils higher than *M* (Eq.1 or 3), which is irrational. This might have been caused by the different time frames or the stimulation of *MFAA* but not of *M* by the AA addition. In any case, numerical $^{15}$N tracing models overcome such inconsistencies, as all gross rates are quantified simultaneously.'.

**III. Nutrient use efficiency** was discussed and Reviewer 3 points out that: **R3**: *'the nitrogen use efficiency model proposed in the present study (NUEFAA) is conceptually different than that by Mooshammer et al. (2014) (NUE)'*. **R3**: *'Furthermore, the authors claim that using gross rates computed by their numerical model (IFAA and MFAA) yields more accurate estimates of microbial nitrogen use efficiency compared to the model by Mooshammer et al. (2014). The authors raise two points: (1) the analytical approach yields less accurate rate estimates (but see comment above), and (2) Mooshammer et al. (2014) use gross N mineralization (M) instead of amino acid mineralization (MFAA). However, the nitrogen use*

*efficiency model proposed in the present study (NUEFAA) is conceptually different than that by Mooshammer et al. (2014) (NUE). The authors estimate here amino acid-N use efficiency (NUEFAA) based on amino acid immobilization and amino acid mineralization, whereas Mooshammer et al. based their model on gross amino acid consumption rate as proxy for microbial organic N uptake, since proteins are the main N-containing compounds in soil and plant litter. Therefore, both NUEFAA and NUE models are conceptually justified and there is no evidence that using MFAA instead of M yields better estimates of microbial NUE. Indeed, estimates of microbial NUE could be improved by including also organic N compounds other than amino acids, which, however, remains a great analytical challenge.'*

**Our reply:** We concur with R3 that there are different thoughts behind the NUE in Mooshammer's paper, the input data differ between the Mooshammer NUE and our NUE$_{FAA}$ and therefore these two methods cannot be directly compared.

- For this reason we have adjusted the manuscript to not focus on this comparison. E.g. in the Abstract, the purpose has been focused to deal with the *Ntrace* model development, and not the NUE calculations, hence we have deleted this sentence: '2) suggest an amino acid N use efficiency (***NUE$_{FAA}$***) for soil microbes, which is a more realistic estimation of soil microbial NUE compared to the NUE estimated by analytical methods.' And modified the following sentence: 'ii) gross FAA mineralization and FAA immobilization rates can be used to develop FAA use efficiency (*NUE$_{FAA}$*), which can reveal microbial N and C limitation.''.

- We have deleted the following sentences from the manuscript introduction: 'The microbial N use efficiency (NUE) representing the balance between immobilization and mineralization, is regulated by the soil organic matter (SOM) quality, e.g. soil C to N ratio (Mooshammer et al. 2014). A soil carbon (C) to N (C/N) ratio of 20 is suggested as a breakpoint where NUE reach a maximum (Mooshammer et al. 2014), as a result of microbial retention of N due to N limitation (at high NUE). Contrastingly, high N mineralization leading to low NUE, results from C limitation (Mooshammer et al., 2014).'; . The text in the introduction concerning NUE is the following: 'Carbon or N limitation of microbes in a soil govern the direction of the soil N flow towards mineralization (N in excess) or immobilization (C in excess) (Robertson and Groffman, 2015).'…'Given the obtained amino acid immobilisation and amino acid mineralization rates, the FAA use efficiency (*NUE$_{FAA}$*) can indicate whether a C or N limitation is occurring.'

- We have deleted this part of methods section: 'results from analytical solution, Mooshammer et al. (2014) calculated ***NUE*** as:

$$NUE = (C_{FAA} - M) / C_{FAA} \qquad (6)$$

Equation 6 implies that gross N mineralization derived from the analytical calculations is solely derived from FAA mineralization.', in order to avoid making the direct comparison.

- Figure 1 has been slightly changed to not have any formulas for NUE shown, the only is now in the text.

- We have deleted figure 6 that directly compared data points from the two ways of calculation. And deleted the analytical solution from Table 2.

- We have modified the discussion by deleting the following section:

    'The observed differences in gross N transformation rates are connected to differences in soil organic matter quality and properties of the microbial biomass (Farrell et al., 2014). The C/N ratios for the two investigated soils were near the breakpoint (C/N ratio of 20) suggested by Mooshammer et al. (2014), at which a change from C limitation to N limitation of the microbial community occur (Fig. 6). By using the gross rates from *Ntrace*, the $NUE_{FAA}$s were 0.57 for Umbrisol and 0.60 for Podzol, which is smaller than expected from the relationship presented by Mooshammer et al. (2014) (Fig. 6). However, the *Ntrace* derived $NUE_{FAA}$s agree with the results from the analytical approach obtained from the longest time step (30 mins to 360 mins), but not for the shorter time steps (Table 2; Fig. 6). For Umbrisol the $NUE_{FAA}$ from the analytical approach (Eq. 6) at the shorter time steps (30 min to 60 min and to 120 min) were higher and fell within the confidence interval from Mooshammer et al. (2014; Fig. 6). We account this to the fact that Eq. (6) uses gross FAA consumption rates quantified by the analytical approach. As it is well understood, this approach provides an overestimation of consumption rates ($C_{FAA}$), due to substrate addition (Schimel, 1996; Di et al., 2000), hereby, the $NUE$ (Eq. 6) will be biased towards high values. The Podzol showed significant input to gross mineralization from other organic N than FAAs therefore, the $NUE$ of Podzol derived from the analytical equation (Eq. 6) (time step 30 min to 120 min) was low. Consequently, $NUE_{FAA}$ is ideally assessed by considering FAA mineralization explicitly (Eq. 5). If the true $NUE_{FAA}$ is lower as we suggest from the *Ntrace* approach, it is likely that a larger portion of FAAs taken up by microbes is subsequently mineralized, than would be suggested from the line in Fig. 6. This challenges the understanding of the shift of soil C limitation to N limitation, however the two investigated soils can neither be termed as N or C limited.'

- We have now added a sentence to the results and discussion about the computed $NUE_{FAA}$: 'The C to N ratio for the two soils near 20, which indicates that the soils are at a tipping point for either C or N limitation, according to the concept from Mooshammer *at al.* (2014; Figure 1). Our result of amino acid nutrient use efficiency ($NUE_{FAA}$) was 0.57 for Umbrisol and 0.60 for Podzol, which point towards a carbon limitation in those soils, as we hypothesized.'.

- Finally we have modified the conclusions point ii) as follows: 'FAA mineralization and FAA immobilization rates can be used for assessing FAA use efficiency ($NUE_{FAA}$) and soil N limitation'.

Finally **R3:** '*Pg 2 L14: From a biological rather than mathematical perspective, I would say that low NUE leads to high N mineralization, and not that high N mineralization leads to low NUE.*'

**Our reply**: We agree, but this sentence was deleted and the topic brought into another sentence, see corrections above.

**IV. Comparing *Ntrace* with analytical calculation R2**: *'Model comparisons. I don't know if there is a 'right' way, but comparing rates from a zero-order analytical model with a mixed kinetic numerical model seems fraught. The attempt to determine an integrated rate for a given time period seems reasonable, but exactly how this was done is not described in much detail. In reality the rates given by the two model types agree quite well.'.* **R1**: *'Abstract page 1, lines 16-17: while stated here and repeatedly throughout the manuscript, that the numerical approach is superior to the analytical method, later in the manuscript (page 8, lines 12-13) it is argued that the numerical model is valid because it produces results for Dson that are similar to the analytical approach. This is contradictory.'* **R1**: *Page 7, line 24: Since both of the presented approaches have their limitations and are biased by the large amount of amended amino acids, I think it is not possible to tell which method is more realistic.'* **R3**: *'how to evaluate which approach yields more realistic rate estimates, which certainly cannot be concluded by simply comparing rates estimated by the two different approaches. I believe that both analytical and numerical models have disadvantages and limitations.… The authors also claim that their numerical model overcomes the problem of high label addition by integrating gross rates over a longer time……. Therefore, I suggest that the authors need to provide some experimental evidence that the numerical model actually overcomes problems such as high label addition. Regarding the comparison between the two different approaches, statistical support should be provided for differences in rates estimated by the two models.'.*

**Our reply:** We concur there were some faults in the phrasing and logic for comparing the obtained rates. In the abstract we have now specified the outcome of the paper as follows: 'In this paper, we present an extended numerical $^{15}$N tracing model *Ntrace*, which incorporates the FAA pool and related N processes in order to provide a more robust and simultaneous quantification of gross production and mineralization rates of FAAs together with gross N mineralization.'. Furthermore, we have removed this sentence from the abstract: 'Due to the short time span, soil disturbance and unnatural high FAA content during the first few hours after the labelling with the traditional $^{15}$N pool dilution experiments, analytical models might overestimate peptide depolymerization rate.'

[Figure]

**Figure III**. Time course of FAA mineralization ($M_{FAA}$) over the 240 hours experimental duration (solid lines) for Umbrisol (red) and Podzol (blue), compared to the average rate (dashed lines; coloured areas ± standard deviation).

Figure III shows with an example from our data, how the *Ntrace* obtained $M_{FAA}$ rate after a longer term incubation is a good estimate of the average rate. However, when calculated at the first hours of the experiment (as done in the analytical model approach) the rates are over estimated.

Regarding the issue of comparing *Ntrace* with analytical calculation we do not think it is contradictory to state that the numerical model is superior, and at the same time validate our model against analytical model results. Any robust numerical method should give the same results as an analytical model (if the same model structure is used), therefore having similar $D_{SON}$ or total mineralization indicates that the numerical method is valid. Analytical models can be useful, and they are correct in terms of the mathematical integration, however numerical models are more robust as they provide estimation of all rates simultaneously and not sequentially. Therefore the output is not an exact solution of the equations but in fact an approximation to the exact mathematical solution. Another advantage is that numerical models have a coherent model concept, which means that we consider processes rather than total production and consumption. Myrold and Tiedje (1986) stated *"The structure of the N cycle makes it amenable to description as a compartmental system. The compartments are the pools of chemically or biologically distinct forms of N and the flows among these pools are the rates of the various N cycle processes."* Gross rates for consuming processes are also more realistic, as we go beyond the phase of rate stimulation (see Fig. II) and can use the model for experiments over longer time, not just 24 hours (which explains the particular differences in those compared to the similar production rates). In numerical models we can also use flexible kinetics as there is no reason why we would expect zero order kinetics for most processes. A more detailed model description of *Ntrace* and comparison with other analytical and numerical tracing models can be found in Rütting and Müller (2007).

**R3:** *'The excessive addition of 15N-AA in the present study likely resulted in biased gross rate estimates, in particular CFAA, and thus also affected the calculation of nitrogen use efficiency (NUE). The stimulation of amino acid mineralization rates due to high 15N-AA label addition, are also likely to have resulted in an overestimation of MFAA rates with the numerical approach (and not only rates estimated by the analytical approach), which would explain the high MFAA rates estimated.'*

**Our reply:** (see also the joint reply to the topic given above) The problem of stimulation of consumption processes will also occur in numerical tracing model, but only at the start of the experiment, as long as the substrate is elevated compared to background. As numerical tracing model integrate rates over longer time periods, this stimulation will be minimized. Indeed, as can be seen in Fig. 5, $M_{FAA}$ was enhanced when estimated for the first 6 hours compared to the entire experimental duration of 240 hours. This points to the fact that the gross consumption rates can in general be quantified unaffected by substrate addition when integrated over longer time periods (but see also the discussion about our high substrate addition). Note also that the AA consumptions quantified by *Ntrace* are indeed lower compared to the analytical rates (Figure II)

**Some additional referee comments were raised:**

**Anonymous referee R1**:

1. *'The introduction should be concluded with concrete testable hypotheses, similar to those presented in the abstract. These hypotheses should be revisited in the discussion.'*

**Our reply**: Thank you for this suggestion, we have at the first introduction section now placed one research question: '…availability, hence our research question is whether the peptide depolymerization and FAA mineralization rates are two equally important steps co-limiting for N availability.', and at the end of introduction rephrased and added two hypothesis: 'In this paper we combine two parallel [15]N tracing experiments, in which soil is separately amended with [15]N labelled ammonium or an amino acid mixture. By splitting the amino acid labelled incubation, two rates (depolymerization rate and amino acid mineralization rate) were assessed from one label. For data analysis, we further developed the numerical [15]N tracing model *Ntrace* (Müller et al., 2007) to explicitly account for FAA turnover, in order to simultaneously quantify gross peptide depolymerization, gross FAA mineralization and total gross N mineralization in forest soils. For our selected mineral soils from Swedish spruce forest, our hypotheses are: 1) FAA mineralization is a major important part of gross N mineralization; 2) due to year-long successful forestry in this area we expect the soil  to be carbon limited rather than N limited.'; In the results and discussion we revisit these hypothesis: '….that amino acid mineralization rate is a major part of the gross N mineralization as hypothesized, and can be considered as a co-limiting step for plant N availability in terrestrial ecosystems.', and: 'The C to N ratio for the two soils near 20, which indicates that the soils are at a tipping point for either C or N limitation, according to the concept from Mooshammer *at al.* (2014; Figure 1). Our result of amino acid nutrient use

efficiency ($NUE_{FAA}$) was 0.57 for Umbrisol and 0.60 for Podzol, which point towards a carbon limitation in those soils, as we hypothesized.'

2. '*Page 2, line 23-24: please state some of these obvious limitations ('These approaches apply analytical calculations (Kirkham and Bartholomew, 1954; Watkins and Barraclough, 1996) handling one flux at the time, which has some obvious limitations (Rütting et al. 2011).')*'

**Our reply:** We specify this as follows in the introduction: ' … limitations: 1. The analytical solutions only provide total consumption and production rates and not the specific processes, 2. analytical solutions only consider zero-order kinetics, 3. the possibility of re-mineralization / re-mobilization limits the experimental work to short time steps, finally 4. with the analytical approach gross rates are sequentially quantified, which does not take into consideration possible interactions; hence, the numerical modelling provides a more coherent framework as the process rates are quantified simultaneously (Rütting et al., 2011).'.

3. '*Page 10 lines 12-14: Especially for the amino acid pool dilution method a longer incubation time of 6H might result in problems with recycled labelled N.*'

**Our reply**: we change our recommendation slightly and delete the brackets: '(e.g. after 6 h)'.

4. '*Figures: Please stat for all figures that include error bars what these are and what the sample size was.*'

**Our reply**: This is now specified in the figure caption for: Figure 2: '..symbols indicate data observation with standard deviation (n = 5; except $^{15}$N fraction of free amino acids: n = 4 at 13 min),… ', Figure 3: '…with standard deviation (n = 5; except $^{15}$N fraction of free amino acids: n = 3 at 13 min)', Figure 4: 'Initial soil content of individual amino acids (µg N-FAA g$^{-1}$ DW soil) indicated as average ± standard error (n = 5).', and Figure 5: '…mineralization rates [ng N g$^{-1}$ h$^{-1}$] indicated as average with deviation (n = 5) for Umbrisol (A) and Podzol (B) ….. total) in [ng N g$^{-1}$ h$^{-1}$] as average with standard deviation (n = 5),….'.

5. '*Figure 1. The second formula for NUE should have IFAA + MFAA in the denominator.*'

**Our reply:** This is correct, however we have deleted both formulas from the figure since we now only work with one formula in the paper.

**Anonymous referee R2**:

1. '*Table 1. It is peculiar that soil ammonium concentration is not provided. If available (and it should be), it should be added. It is important for the reader to know how the amount of 15N-label added compares to the natural background.*'

**Our reply:** We have provided the ammonium data. NH$_4$ concentration in µg g$^{-1}$ DW soil (average and standard error) was for Podzol: 1.4 ± 0. 6 and for Umbrisol: 1.1 ± 0.9. These data are added to Table 1.

2. '*Units. Although maybe not the best, most 15N tracer studies provide concentrations and rates in terms of mass of N (e.g. mg, ug) rather than moles N. I would suggest tables and figures be converted to mass of N to make the data easily comparable to previous studies. Also rates are most often 'per day' rather than 'per hour'.*'

**Our reply:** We have changed the units to $ngN\ g^{-1}\ h^{-1}$ in Figure 2, 3 and 5 and Table 2, and in the figure captions.

3. '*...types agree quite well (Table 2) except for CFAA, which leads me to question the validity of Eqn 4. I was too lazy too check Barracloughs derviation, but it makes me wonder if there is a flaw in Eqn. 4. Could the averaging to get the a' values be a factor?*'

**Our reply:** Originally the equation was developed by Watkins and Barraclough (1996) for plant residues, and in that case the added plant material had a constant $^{15}N$ excess (as no new residue was formed) that was used for the calculation. However, as in the case of amino acids the $^{15}N$ excess changes over time (due to production of AAs), we rather use the average $^{15}N$ excess of the two points. This averaging is similar to what Huygens et al. (2008) did for DNRA quantification (which is based on a similar thinking, see supplementary material).

Specific corrections

4. '*P1 l.8 use commas to set off the 'such as' phrase. I also wonder if monomers is too limiting ? It was the authors measure in this research, however, others (Farrell et al. I believe) have shown that small oligopeptides are preferred over amino acids.*'

**Our reply:** We have modified the first sentence as follows: 'Depolymerization of soil organic matter, such as proteins and (oligo-) peptides into monomers (e.g. amino acids) is currently considered to be the rate-limiting step for nitrogen (N) availability in terrestrial ecosystems.'

5. '*P2 l 15: A good classical reference that would fit here is Jansson and Persson 1982. Mineralization and immobilization of soil nitrogen p 229-252. In Steveson (ed.) Nitrogen in agricultural soils. ASA Madison, WI.*'

**Our reply:** We were not able to find this reference, but we have added this one instead: Robertson and Groffman 2015;. 'Carbon or N limitation of microbes in a soil governs the direction of the soil N flow towards mineralization (N in excess) or immobilization (C in excess) (Robertson and Groffman, 2015).' For the reference list: Robertson, G.P. and Groffman, P.M., Nitrogen transformations. Cpt. 14 in Soil microbiology, ecology, and biochemistry. Ed. Paul, E.A. 4th edition, Academic Press, Elsevier. 2015.

6. '*P2 l 15: delete 'to address', P 2 l 19 'thereby' rather than 'hereby'*'

**Our reply:** Corrected accordingly.

7. '*P 2 l 31: define NUEFAA.*'

**Our reply**: The sentence was deleted.

8. *'P3 l 9: parantheses around the year. P3 l 15 the #th# by dazes can be deleted. P3 l 24: capitalize Laboratories.'*

**Our reply:** Corrected accordingly.

9. *'P4 l 4: provide the rationale for the calcium sulfate / formaldehyde extraction. It is not a standard method that readers will know.'*

**Our reply**: We have added this sentence to the text: 'The $CaSO_4$ was selected because deionized water alone lyses microbial cells, thereby releasing a large flux of amino acids from the cells. Formaldehyde was used in order to inhibit microbial consumption or activity during the shaking time.'

10. *'P 4 l 25: it is more typical to oven dry mineral soils at 105 C; 75 C is more normal for plant tissue (or organic soil horizons).'*

**Our reply**: We agree that this is classical, but as we wanted to analyse the same soil samples for total C and N, we did not want to risk losing volatile organic matter (C and N) during the drying procedure, for this reason we prefer 75 deg. C.

11. *'P 5 l 5 The Andresen ref is inappropriate here as the equations come directly from the original Kirkham and Bartholomew paper.'*

**Our reply:** we have removed the reference.

12. *'P5 l 18 Are you sure it is logarithmic, or was it exponential ?'*

**Our reply**: the functions are indeed logarithmic (example: M, *mg N kg$^{-1}$ hr$^{-1}$* = -0.023ln(x, *hr*) + 0.169).

13. *'P 6 l 6: delete the comma after 'could'.'*

**Our reply:** we have removed it.

14. *'P6 l 8 I don't think the pool is 'infinite', it is large and changes imperceptibly during the short incubation period'*

**Our reply:** We have rephrased this: 'The N transformations were either implemented as zero-order kinetics for large substrate pools that is constant in size during the incubation ($D_{SON}$ and $M_{SON}$) or first-order kinetics for finite pools ($M_{FAA}$, $I_{FAA}$ and $I_{NH4}$).'

15. *'P 6 l 28 add % after '22'.'*

**Our reply:** Corrected accordingly. .

16. *'P 7 l 15 this seems like a throw away sentence as it does not really lead to greater understanding of the results.'*

**Our reply:** We have rephrased the sentences: 'Numerical tracing models represent robust methods to assess gross transformation rates, as all data points from the two isotope label experiments and all observed time steps are included. To our knowledge, quantification of

total gross FAA mineralization and peptide depolymerization rates had not been done by numerical tracing models.'.

17. '*P 7 l 18 problem of additions stimulating processes is as true regardless of what approach one uses to analyse the data. Now, it is true that if the model uses first-order kinetics, then this "mass dependency" is accounted for to some degree, but one can incorporate these kinetics in the analytical model (see "Case 2" in the rarely quoted 1955 paper by Kirkham and Bartholomew).*'

**Our reply:** Indeed, Kirkham and Bartholomew developed in their 1955 paper an analytical model using first-order kinetics for $NH_4$ consumption. However, this model assumes that all $NH_4$ is immobilized and none is nitrified, i.e. that a closed N cycling between SON and $NH_4$ exists. This will almost never be the case, for which reason that first-order analytical model is almost never applicable.

18. '*P 7 l 16 correct to 'assess'.*'

**Our reply:** This is done.

19. '*P 8 l 6 Is "irrational" the best word? The result is illogical, but that raises a question*

*as to whether there is a flaw in the logic behind equation 4?*'

**Our reply:** The equation 4 is indeed correct, see reply to R1, but the mentioned sentence was deleted during responding to R1.

20. '*P 10 l 5 change to 'points'.*'

**Our reply:** This is done.

21. '*P 10 l 26 Chitin is not an amino acid, it is an amino sugar polymer.*'

**Our reply:** this is now specified: 'Another outlook is that depolymerization rates of polymers other than amino acids (such as amino sugar polymers) are potentially an important part of the total depolymerization.'.

22. '*Capitalization of titles is inconsistent (particularly older references)*'

**Our reply:** This is corrected accordingly.

**Anonymous referee R3**:

1. '*In Figure 2d and 3d, it becomes also evident that the 15N-AA label was not homogenously distributed, otherwise there would have not been an increase in 15N/14N in the first time interval. Of course, for this time interval, it is not possible to calculate a rate using the analytical approach. The weak dilution of the 15N-AA label during the first time intervals was not due to low depolymerization rates, as suggest by the authors, but rather likely due to high 15N-labelled substrate addition, which resulted in an enrichment of the FAA pool of 60-70 at%.*'

**Our reply:** We assign this slight (insignificant) increase in [15]N rather to heterogeneity of the soils, even though that homogenized samples were used. The high [15]N enrichment should rather increase the sensitivity to detect small production, as low inflow of unlabelled material should lead to a more visible dilution of highly enriched pools compared to low enriched pool. We do not see how the high [15]N enrichment should lead to weak dilution. Therefore, we still conclude that the low dilution is indeed due to low depolymerization rate.

2.*'Equations of the analytical approach (Eq. 1, 2 and 3) are partially wrong: equation 1 for production rate has a mistake in the numerator of the second term; equation 2 for the consumption rate is actually the production rate; and, equation 3, for the case when there is no change in concentration over time, also seems wrong, at least when compared to their former work (Andresen et al. 2015).'*

**Our reply:** All three equations are correct. However, equations are slightly different presented compared to Andersen et al. (2015). While in the former paper we presented equations with the [15]N excess fraction (a) as input, here we use the equations analogous to the original Kirkham and Bartholomew paper using the [15]N excess amount (H) as input.

3. *'Furthermore, the authors used different units: μg N/g, μg FAA/g, μmol N/g, nmol N/g. In some instances, it is not even stated whether it refers to N or FAA (Table 1, Table 2). I suggest using consistent units throughout the manuscript. As I understood it, the initial FAA concentrations presented in Table 1 should correspond to the sum of FAA presented in Figure 4. For Podzol, Table 1 shows 1.3 μg, whereas in Figure 4 the values roughly sum up to 3 μg. In turn, for Umbrisol, both Table 1 and Figure 4 seems consistent: 7.7 μg in Table 1, as well as roughly 7.7 in Figure 4.'*

**Our reply:** We have now standardized this as much as possible: in Table 1 and Figure 4 the FAA unit is now both consistently μg N/g from FAAs, and specified this in the Table heading: 'and total free amino acid content (FAA in μg N g$^{-1}$DW).' And Figure caption: 'Figure 4. Initial soil content of individual amino acids (μg N-FAA g$^{-1}$ DW soil) indicated as average ± standard error (n = 5). ', and axis title: 'Free amino acid content (μg N g$^{-1}$ )'.

4. *'Pg 1 L14: Comma is missing before which; Pg 1 L16: Delete "2)".'*

**Our reply**: Corrected accordingly.

5. *'Pg 2 L13: correct to "reaches".'*

**Our reply**: this whole sentence complex was removed, see other reply.

6. *'Pg 2 L23-24: The authors should state what the obvious limitations are.'*

**Our reply**: See reply above to R1.

7. *'Pg 4 L1-2: For N mineralization, 13 min for equilibration between added and native*

*ammonium is quite short.'*

**Our reply**: The important information from the initial extraction is to what extend the $NH_4$ pool was labelled. The immobilization starts as soon as the $NH_4^+$ is added, therefore we aimed to keep this time as short as possible. Waiting too long between addition and first extraction leads to an underestimation of $NH_4^+$. Even though that a homogenous label distribution might

not have been achieved at that point (if it ever can), we do not see that this could be a major issue for the quantifications.

8. *'Pg 4 L3-4: Why were FAAs also extracted with 1 M KCL?'*

**Our reply**: The sample labelled with AAs was split in two: one sample was extracted with $CaSO_4$ solution and one was extracted with KCl. This was in order to measure $^{15}N\text{-}NH_4$ in the KCl sample. For clarification, the following text is added to the end of this section: 'The KCl extract was made for $^{15}N\text{-}NH_4$ analyses.'.

9. *'Pg 4: The authors used the protocol developed by Wanek et al. (2010). However, the*

*authors do not even once cite this work in Materials & Methods.'*

**Our reply**: The reference to Wanek et al. 2010 is now appropriately added to the methods section.

10. *'Pg 4 L16: The reference is wrong. It should be Husek, 1991.'*

**Our reply**: The mistake is corrected.

11. *'Pg 4 L18-23: The authors should explain why some amino acids have the same m/z:*

*Alanine and Glycine (m/z 116/117); Leucine, Serine, Isoleucine and Threonine (m/z*

*158/159); Proline and Aspartic acid (m/z 142/143)'*

**Our reply**:  We thank the reviewer for nothing this error, the m/z given in the original manuscript were wrong, and is corrected now: 'Alanine (Ala m/z: 116/117), Glycine (Gly m/z: 102/103), Valine (Val m/z: 144/145), Leucine and Isoleucine (Leu and Ile m/z: 158/159), Serine (Ser m/z: 131/132), Threonine (Thr m/z: 146/147), Proline  (Pro m/z: 142/143), Aspartic acid (Asp m/z: 188/189), Asparagine (Asn m/z: 141/143), Methionine (Met m/z: 249/250), Glutamic acid (Glu m/z: 202/203), Phenylalanine (Phe m/z: 192/193), Lysine (Lys m/z: 156/157), Tyrosine (Tyr m/z: 280/281) and Tryptophan (Trp m/z: 130/131).' (This replaces the following text in the methods section: 'Alanine and Glycine (Ala and Gly; m/z: 116/117), Valine (Val; m/z: 144/145), Leucine, Serine, Isoleucine and Threonine (Leu, Ser, Ile and Thr; m/z: 158/159), Proline and Aspartic acid (Pro and Asp; m/z: 142/143), Asparagine (Asn, m/z: 188/189), Methionine (Met, m/z: 249/250), Glutamic acid (Glu; m/z: 202/203), Phenylalanine (Phe; m/z: 192/193), Lysine (Lys; m/z:156/157), Tyrosine (Tyr; m/z: 280/281) and Tryptophan (Trp; m/z: 130/131)'). Only for Leucine and isoleucine (isomers) the same m/z was used (they are well separated chromatographically). The selected ion fragments are similar to those selected in Wanek et al. 2010, and were selected as N containing ion fragment with high intensity. In most cases the selected ion correspond to fragment resulting from the loss of $^\bullet CO_2CH_2CH_3$, though for some (methionine, lysine, threonine, histidine, tryptophan) another fragment had to be selected (due to low intensity or interfering fragments).  We believe that giving the structure of the used ion fragments is beyond the scope of this paper.

12. *'Pg 5 L5 and L15: The original references (Kirkham and Bartholomew, 1954; Watkins*

*and Barraclough, 1996) are sufficient. Delete reference Andresen et al., 2015.'*

**Our reply**: Done in both places.

13. *'Pg 5 L13: Specify that it is excess 15N abundance.'*

**Our reply**: We have corrected to: 'Excess $^{15}$N content' (but this is not the same as abundance).

14. *'Pg 6 L15: I do not always see a good fit of the model in Figure 2 and 3. For example in Figure 3b, the fit of the model for ammonium concentration seems not to fit the experimental data.'*

**Our reply:** Indeed, the model does not lie within the uncertainty of all individual data points. However, importantly the overall performance of the model is quite well given that (1) the overall trends are well represented and (2) that the majority of data points are fitted by the model (particularly for data in Fig. 2). For the data in Fig. 3 (Podzol) the main challenge was the very low dilution of $^{15}$N in the FAA, which will also affect how good the other pools are represented by the model. To achieve a better model fit in future studies, we suggest in the paper to have a later measurement point in the $^{15}$N-AA labelling treatment to better follow the NH$_4$ pool and its $^{15}$N enrichment (section 3.4 Suggested improvements of the laboratory method).

15. *'Pg 8 L18-19: In Wanek et al. (2010) the samples were plant litter and not organic soil.'*

**Our reply**: We have corrected the mistake and rephrased: 'The ratio of total gross N mineralization (*M*) to peptide depolymerization ($D_{SON}$) rate ranges from 5 to 25 % in both organic soils and plant litter, based on analytical calculations (Wanek et al., 2010; Wild et al., 2015).'

16. *'Figure 1 B: Equation is wrong.'*

**Our reply**: Equations were removed, see reply earlier to the common question of NUE.

17. *'Table 1: The initial soil ammonium concentrations of both soils should be stated. There*

*is no need to say that the C:N ratio refers to dry soil.'*

**Our reply**: The ammonium data is added, se reply to R2. And reference to 'dry' is removed from the table heading.

18. *'Figure 4: Typo on the y-axis.'*

**Our reply**: Corrected.

[revised manuscript text omitted]